# CoDA: Agentic Systems for Collaborative Data Visualization

**Zichen Chen**[*]
UC Santa Barbara
zichen_chen@ucsb.edu

**Jiefeng Chen**
Google Cloud AI Research
jiefengc@google.com

**Sercan Ö. Arık**
Google
soarik@google.com

**Misha Sra**
UC Santa Barbara
sra@cs.ucsb.edu

**Tomas Pfister & Jinsung Yoon**
Google Cloud AI Research
{tpfister,jinsungyoon}@google.com

https://coda-agent.github.io/CoDA/

## Abstract

Deep research has revolutionized data analysis, yet data scientists still devote substantial time to manually crafting visualizations, highlighting the need for robust automation from natural language queries. However, current systems struggle with complex datasets containing multiple files and iterative refinement. Existing approaches, including simple single- or multi-agent systems, often oversimplify the task, focusing on initial query parsing while failing to robustly manage data complexity, code errors, or final visualization quality. In this paper, we reframe this challenge as a collaborative multi-agent problem. We introduce CoDA, a multi-agent system that employs specialized LLM agents for metadata analysis, task planning, code generation, and self-reflection. We formalize this pipeline, demonstrating how metadata-focused analysis bypasses token limits and quality-driven refinement ensures robustness. Extensive evaluations show CoDA achieves substantial gains in the overall score, outperforming competitive baselines by up to 41.5%. This work demonstrates that the future of visualization automation lies not in isolated code generation but in integrated, collaborative agentic workflows.

## 1 Introduction

Data visualization plays an important role in business intelligence, data science and decision-making, enabling professionals to uncover insights from complex datasets through intuitive graphical representations (Ramesh & Rajabiyazdi, 2024; Gahar et al., 2024; Jambor, 2024; Beschi et al., 2025; Rogers et al., 2024). In practice, data analysts might spend over two-thirds of their time on low-level data preparation and visualization tasks, often iterating manually to achieve clarity, accuracy, and aesthetic appeal (Lai et al., 2025; Rezig et al., 2021; Lee et al., 2021). This "unseen tax" diverts focus from insight generation, highlighting the critical need for automated systems that can transform natural language queries and complex data into effective visualizations (Wu et al., 2024; Chen et al., 2024; Wang & Crespo-Quinones, 2023). With the rise of large language models (LLMs) (Naveed et al., 2025; Achiam et al., 2023; Team et al., 2024; Comanici et al., 2025), there is immense potential to automate this pipeline. However, realizing this potential requires addressing core challenges: (1) handling large datasets, (2) coordinating diverse expertise (e.g., linguistics, statistics, design), and (3) incorporating iterative feedback to refine outputs against real-world complexities like messy multi-file data and complex visualization needs.

Current approaches to automate visualization suffer from various limitations. Traditional rule-based systems, such as Voyager (Wongsuphasawat et al., 2017; 2016) and Draco (Yang et al., 2023), formalize design knowledge as constraints but remain confined to predefined templates, struggling with natural language queries or diverse data patterns (Wu et al., 2024; Hoque & Islam, 2025). LLM-based methods, like CoML4VIS (Chen et al., 2024), leverage chain-of-thought prompting to generate visualizations (Comanici et al., 2025), but often ingest raw data directly, risking token limit violations, hallucinations, and failures on multi-source data (Bai et al., 2024; Chen et al., 2024). Multi-agent

---

[1]This work was done while Zichen was a research intern at Google Cloud AI Research.

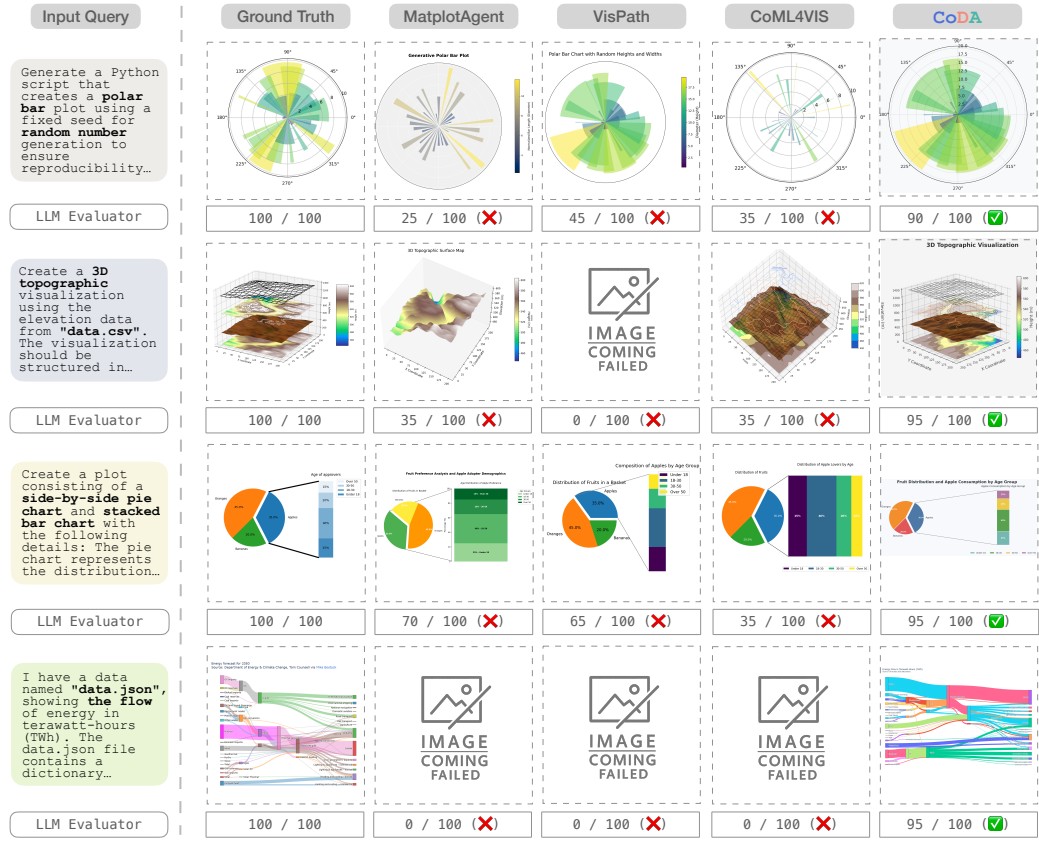

Figure 1: Qualitative comparison of visualizations generated by baselines (*MatplotAgent*, *VisPath*, *CoML4VIS*) and CoDA. Provided with a natural language query and data files (if has), models produce code to create plots. CoDA yields outputs that more faithfully capture complex patterns, chart types, and aesthetics, while baselines often fail on ambiguity, 3D structures, or multi-source integration.

frameworks, such as VisPath and MatplotAgent, introduce a collaboration system to generate plot code but lack metadata-focused analysis, leading to brittle data processing and poor robustness under iterative refinement (Seo et al., 2025; Yang et al., 2024b). We argue these issues stem from a common limitation in current agentic visualization systems: they concentrate reasoning and coordination on initial query parsing, which proves insufficient for handling complex data environments (e.g., multiple and large files), code errors, and iterative refinement. This design limits their ability to adapt to unexpected data challenges.

To address these challenges and limitations, we propose CoDA (Collaborative Data-visualization Agents), a multi-agent system that advances visualization automation through a self-evolving pipeline where agents specialize in understanding, planning, generating, and reflecting. By analyzing metadata schemas and statistics without raw data file uploads, we circumvent the context window limits of LLMs; specialized agents enhance domain reasoning; and image-based evaluation assesses output quality from a human-centric perspective. This builds a robust framework for complex, iterative, and multi-source agentic visualizations, where agents collaborate deeply to ensure visualization quality. The key contributions of this work are as follows:

- We propose CoDA, an extensible framework with specialized agents for metadata analysis, task planning, code generation and debugging, and self-reflection, enabling robust handling of complex data and visualization needs (see Figure 1 and Appendix B for qualitative analyses).
- Extensive experiments on MatplotBench and Qwen Code Interpreter benchmarks yield substantial gains in the overall score over strong baselines such as *MatplotAgent*, *VisPath*, and *CoML4VIS*, with maximum improvements of 24.5%, 41.5%, and 26.5%, respectively. Further-

more, CoDA significantly outperforms competitive baselines on the DA-Code benchmark, which features complex, real-world software engineering scenarios.

- A comprehensive ablation study validates the necessity of CoDA's core components. Results demonstrate that self-evolution, the global TODO list, and the example search agent each provide a statistically significant positive impact on overall performance.

## 2 RELATED WORK

**Natural Language to Visualization (NL2Vis).**   NL2Vis approaches have revolutionized data exploration in data science by allowing users to articulate queries in natural language and receive target visualizations (Wang & Crespo-Quinones, 2023; Shen et al., 2022; Wu et al., 2024), thereby accelerating initial data scouting and ad-hoc reporting (Voigt et al., 2022). A survey on natural language generation for visualizations provides a taxonomy of techniques and highlights the challenges in ensuring coherence and fidelity to underlying data information (Hoque & Islam, 2025). Many methods have formalized this evaluation landscape (Chen et al., 2024; Ouyang et al., 2025; Bai et al., 2025; Shin et al., 2025) and use chain-of-thought prompting strategies to enhance LLM accuracy on single-table tasks (Liu et al., 2025). These tools are important for data scientists navigating the exploratory phase (Zhang et al., 2025; Chen et al., 2025), but they exhibit gaps in LLM reasoning under ambiguity or multi-source data environments (Zhu et al., 2025; Davila et al., 2025). Empirical evaluations of LLMs in visualization generation reveal shortcomings in CoT-based methods, emphasizing the need for robust handling of abstract and multifaceted queries in decision-making workflows (Khan et al., 2025), motivating our shift toward autonomous multi-agent teams.

**Agentic Visualization Systems.**   Agentic systems mark a paradigm shift in visualization for data science, framing it as a distributed problem-solving process among AI agents that mirror collaborative human co-workers (Sapkota et al., 2025; Tran et al., 2025; Wolter et al., 2025; Xu et al., 2025). Goswami et al. (2025) and Zhang & Elhamod (2025) exemplify this by deploying multi-agent LLM frameworks for autonomous professional visualization, streamlining visual analytics from raw, unstructured data. Yang et al. (2024b) introduce a multi-step reasoning agent framework for scientific plotting, empowering data scientists with code-free handling of complex visualizations. Seo et al. (2025) enhance this through multi-path reasoning and feedback optimization for code synthesis from natural language. Efforts to extract agent-based design patterns from visualization systems provide a blueprint for balancing autonomy with human oversight, laying groundwork for scalable tools in collaborative data environments (Dhanoa et al., 2025). These agentic systems help compress hours of manual labor in data science (Moss, 2025; Gridach et al., 2025). However, they commonly take shortcuts, focusing adaptations on initial planning stages without persistent reflection (Wang et al., 2025; Sapkota et al., 2025). This shallow agentic alignment contributes to vulnerabilities in complex scenarios (Cemri et al., 2025; Tian et al., 2025). Our proposed multi-agent system counters this by enforcing deeper collaboration, via specialized agents for planning, building, criticism, and reflection, to yield robust narratives.

## 3 METHOD

In this section, we formalize the collaborative multi-agent paradigm for data visualization and introduce CoDA. We begin by outlining the key design principles that support agentic visualization systems, drawing parallels to human collaborative workflows in data analysis and plotting. We then describe CoDA's architecture, including the specialized agents and their interactions, and explain how this framework addresses core challenges in automated visualization.

### 3.1 THE COLLABORATIVE MULTI-AGENT PARADIGM

Conventional visualization systems, whether rule-based or LLM-driven (Khan et al., 2025; Zhu et al., 2025; Hutchinson et al., 2024; Shin et al., 2025), typically treat visualization as a monolithic, single-pass process of parsing a query, ingesting data, and generating code. This leads to unstable performance on complex queries involving multi-file datasets, ambiguous requirements, or iterative refinements. We reframe visualization as a collaborative problem-solving endeavor. Our approach employs a team of specialized LLM agents, each with a distinct professional persona, that uses

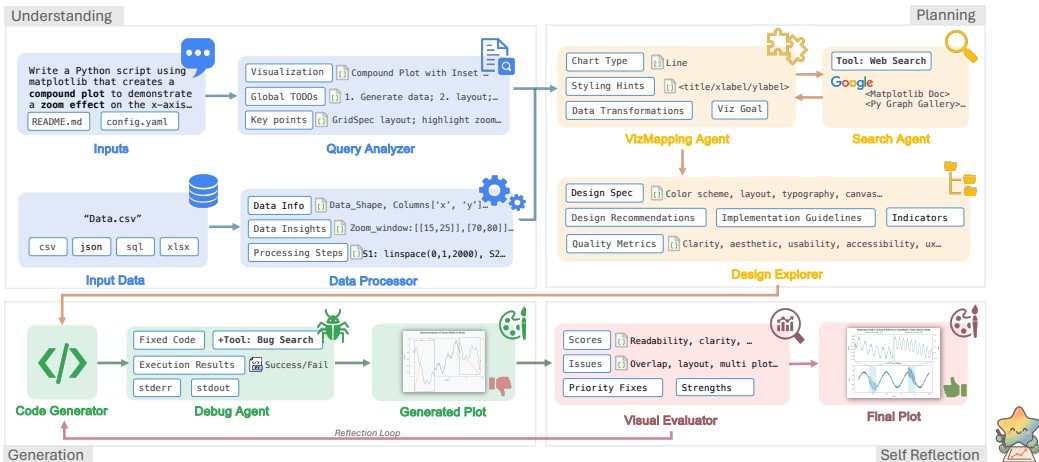

Figure 2: Overview of the **CoDA** framework for agentic data visualization. The workflow decomposes natural language queries into modular phases: **Understanding** (query intent and data metadata extraction), **Planning** (example code search, visual mappings, and design optimization), **Generation** (code generation and debugging), and **Self-Reflection** (quality evaluation with feedback loops for self-reflection refinement).

structured communication and quality-driven feedback loops to decompose queries, process data, and iteratively refine outputs.

Inspired by multi-agent systems in software engineering (Yang et al., 2024a) and interactive reasoning (Yao et al., 2022), this paradigm leverages the emergent capabilities of LLMs to simulate division of labor. Each agent is designed to focus on a well-defined expertise area, such as metadata extraction or code debugging, while communicating via a shared state to adapt dynamically. This not only mitigates token limits by avoiding raw data ingestion but also enhances robustness through reflection and error correction, mirroring how data analysts collaborate to refine insights. Key principles guiding this approach include:

**Specialization for Depth:** Assign agents to distinct roles (e.g., planning vs. execution) to deepen reasoning without overwhelming a single model.

**Metadata-Centric Preprocessing:** Summarize data structures upfront to inform downstream decisions, bypassing the need for full data loading.

**Iterative Reflection:** Incorporate human-like evaluation of outputs (e.g., via image analysis) to detect and correct issues like visual clutter or factual inaccuracies.

**Modular Extensibility:** Design agents as interchangeable modules, allowing integration of new tools or models for evolving tasks.

By unifying query understanding, data handling, code generation, and quality assurance into a self-reflection workflow, this approach transforms visualization from isolated code generation into a resilient, adaptive process. We demonstrate its efficacy through **CoDA**, which operationalizes these principles for real-world benchmarks.

### 3.2    CoDA: COLLABORATIVE DATA VISUALIZATION AGENTS

**CoDA** instantiates the collaborative paradigm as a multi-agent system that takes a natural language query and data files as input, producing a refined visualization as output. Figure 2 provides a high-level overview and Table 1 summarizes the inputs and outputs of different agents in the workflow. Full agent prompts and I/O are shown in Appendix F.

The workflow proceeds as follows, with iterative refinement triggered by quality assessments: The Query Analyzer interprets the query (e.g., *"Plot sales trends by region"*) to extract intent, decomposes it into a global TODO list (e.g., data filtering, aggregation, chart selection), and generates guidelines

Table 1: Inputs and outputs of different agents in the proposed CoDA framework.

| Agent Name | Inputs | Outputs |
|---|---|---|
| Query Analyzer | Query; meta_data (e.g., `README.md`) | Visualization types; key points for plotting; global TODO list. |
| Data Processor | Data inputs | Data info (e.g., `shapes`, `columns`); insights (e.g., `aggregations_needed`); processing steps; visualization hints. |
| VizMapping Agent | Query; query analyzer; data processor results | Chart types; styling hints; transformations (e.g., `aggregations`, `filters`); visualization goals. |
| Search Agent | Visualization types; chart types | Code examples. |
| Design Explorer | Query analyzer; data processor results | Design specifics (e.g., `color_scheme`, `layout`); implementation guidelines; quality metrics; design recommendations; alternatives; success indicators. |
| Code Generator | Design explorer; data processor; search agent; visual evaluator | Generated code; code quality score; dependencies; documentation. |
| Debug Agent | Code generator results | Debugging outputs/errors; web-searched fixes; corrected code; execution results; output file. |
| Visual Evaluator | Output file; query; analyzer; processor results | Scores (e.g., `overall_score`, `readability`); strengths; issues; priority fixes; modifications; recommendations. |

for downstream agents. The Data Processor extracts metadata summaries (schemas, statistics, patterns) from data files using lightweight tools like pandas, avoiding token limits while identifying insights and potential transformations. The VizMapping Agent maps query semantics to visualization primitives, selects appropriate chart types (e.g., line chart for trends), defines data-to-visual bindings, and validates compatibility based on metadata. This agent ensures insightful outputs that adapt to data complexities without raw ingestion. The Search Agent (as a tool) retrieves relevant code examples from plotting libraries (e.g., Matplotlib) to inspire generation, formulates search queries and ranks results by relevance. The Design Explorer generates content and aesthetic concepts, optimizes elements like colors and layout, and evaluates designs for user experience. The Code Generator synthesizes executable Python code integrating specifications, ensuring best practices and documentation. The Debug Agent executes code with timeouts, diagnoses errors (e.g., via searched solutions), applies fixes (potentially via searched solutions), and outputs results like visualization images. The Visual Evaluator assesses the output image across multi-dimensional quality metrics (clarity, accuracy, aesthetics, layout, correctness), verifying TODO completion and suggesting refinements.

Agents exchange structured messages through a shared memory buffer, propagating context (e.g., metadata informs planning, plans guide code). Feedback loops trigger self-reflection: If quality scores (from evaluation) are below thresholds, issues are routed back to upstream agents (e.g., low aesthetics back to the Design Explorer). The system halts when quality converges or reflection limits are reached. CoDA's modular design promotes scalability—agents can be parallelized or extended (e.g., for scientific plotting)—and robustness through quality-driven halting (e.g., stopping when scores exceed thresholds). In experiments (Section 4), this yields substantial gains over baselines, validating the value of this agentic approach in visualization automation.

## 4 EXPERIMENTS

We evaluate CoDA's ability to generate high-quality visualizations from natural language by testing it on a diverse set of visualization benchmarks. We compare CoDA against state-of-the-art baselines using standardized metrics that capture execution reliability, visualization correctness, and overall task success. All experiments are conducted using `gemini-2.5-pro` as the underlying LLM, with a maximum of 3 refinement iterations and a quality threshold of $\theta_q = 0.85$ for halting.

## 4.1 BENCHMARKS

We select benchmarks that span varying levels of complexity in natural language to visualization tasks, including handling diverse data types, chart styles, and user intents. The primary datasets are:

**Qwen Code Interpreter Benchmark (Visualization)** (Yang et al., 2025): This subset focuses on visualization tasks within a code interpretation framework, with 163 examples emphasizing numerical data handling, pattern recognition, and code synthesis for plots. It tests robustness to ambiguous queries and data inconsistencies.

**MatplotBench** (Yang et al., 2024b): A comprehensive benchmark for matplotlib-based visualization generation, comprising 100 queries across domains such as time-series analysis, categorical comparisons, and multi-dimensional plotting. Queries require interpreting user intent, selecting appropriate chart types, and ensuring visual clarity.

These benchmarks represent mid-to-high complexity tasks suitable for evaluating agentic systems in controlled environments. Additionally, we separately evaluate on the more challenging **DA-Code** benchmark (Huang et al., 2024), which involves repository-based software engineering tasks with visualization components. Unlike the above, DA-Code (vis) requires navigating codebases, integrating visualizations into broader workflows, and handling domain-specific constraints (e.g., performance optimization in plots). It comprises 78 tasks and is treated independently due to its elevated difficulty and shift toward SWE-oriented reasoning.

## 4.2 BASELINES

We compare CoDA against recent visualization-specific methods that leverage LLMs for code generation and refinement:

**MatplotAgent** (Yang et al., 2024b): A single-agent system focused on matplotlib code synthesis from queries, with basic error handling but limited multi-step planning.

**VisPath** (Seo et al., 2025): An approach based on multiple solution planning that decomposes visualization tasks into sequential steps, emphasizing path optimization for chart mapping.

**CoML4VIS** (Chen et al., 2024): A workflow-centric framework that follows a structured pipeline to generate visualizations, incorporating table descriptions and code execution.

All baselines use the same `gemini-2.5-pro` backbone for fair comparison, and we follow their papers to set up the parameters (e.g., iteration limits).

## 4.3 EVALUATION METRICS

To provide a multi-dimensional assessment, we define three key metrics that capture execution reliability, visualization quality, and overall task success:

**Execution Pass Rate (EPR):** The proportion of queries for which the generated Python code executes without runtime errors, capturing basic syntactic and dependency reliability. Formally, $EPR = \frac{|q \in Q : \text{exec}(c_q) = \text{success}|}{|Q|}$, where $c_q$ is the code for query $q \in Q$.

**Visualization Success Rate (VSR):** The average score reflecting the quality of rendered visualizations among executable codes, where higher scores indicate closer alignment with intended representations (e.g., accurate data mappings). Formally, $VSR = \frac{\sum_{q \in Q_{\text{exec}}} s_v(q)}{|Q_{\text{exec}}|}$, where $s_v(q)$ is the LLM-evaluated visualization score for query $q$, and $Q_{\text{exec}}$ is the set of queries with successful execution. On a binary-scored benchmark (e.g., Qwen Code Interpreter), VSR reduces to the proportion of correct visualizations among executable cases.

**Overall Score (OS):** The overall score reflects the average of code and visualization quality scores and provides a holistic view of system effectiveness. Formally, $OS = \frac{\sum_{q \in Q} \text{avg}(s_c(q), s_v(q))}{|Q|}$, where $s_c(q)$ is the code quality score and $s_v(q)$ is as defined above.

Additional technical details on the judging prompts and model setup are provided in Appendix E. We emphasize that all reported metrics are computed against benchmark ground truth; the Visual

Table 2: Performance comparison against three baselines on the MatplotBench and Qwen Code Interpreter benchmarks. All baselines utilize `gemini-2.5-pro` as the base LLMs.

| Method | MatplotBench | | | Qwen Code Interpreter | | |
|---|---|---|---|---|---|---|
| | EPR (%) ↑ | VSR (%) ↑ | OS (%) ↑ | EPR (%) ↑ | VSR (%) ↑ | OS (%) ↑ |
| MatplotAgent | 97.0 | 56.7 | 55.0 | 81.6 | 79.7 | 65.0 |
| VisPath | 75.0 | 37.3 | 38.0 | 86.5 | 94.3 | 81.6 |
| CoML4VIS | 76.0 | 69.7 | 53.0 | 87.1 | 90.9 | 79.1 |
| CoDA (Ours) | **99.0** | **79.8** | **79.5** | **93.3** | **95.4** | **89.0** |

Table 3: Comparison of CoDA against DS-STAR (Nam et al., 2025) and DA-Agent (Huang et al., 2024) on the DA-Code benchmark, where DA-Agent is powered by various LLMs including `gemini-2.5-pro`, `gpt-4o`, `gpt-4`, and `deepseek-coder`. Green shading marks the best within each group.

| Metric | CoDA (Ours) | DS-STAR | DA-Agent (backbone LLM) | | | |
|---|---|---|---|---|---|---|
| | Gemini-2.5-pro | Gemini-2.5-pro | Gemini-2.5-pro | GPT-4o | GPT-4 | Deepseek-Coder |
| Overall Score (%) | **39.0** | 20.5 | **19.23** | 17.0 | 16.0 | 11.0 |

Evaluator within CoDA serves solely as an internal guidance signal for self-refinement and is not used as an evaluation metric.

## 4.4 MAIN RESULTS

Table 2 presents the main results on MatplotBench and the Qwen Code Interpreter Benchmark (vis). CoDA outperforms all baselines across metrics, achieving substantial gains in OS of 24.5% on MatplotBench and 7.4% on Qwen over the best alternative, demonstrating superior handling of complex queries through agent collaboration and feedback loops. The high EPR reflects robust code generation, while VSR highlights effective refinement in visualization quality.

## 4.5 RESULTS ON DA-CODE BENCHMARK

In this evaluation, we extend CoDA to more complex, real-world SWE scenarios where visualizations are embedded within broader codebases. Table 3 summarizes these findings, revealing CoDA's score of 39.0%, a 19.77% absolute gain over DA-Agent with `gemini-2.5-pro`, the strongest baseline. Compared to DS-STAR (Nam et al., 2025), a recent state-of-the-art data-science agent system, CoDA achieves nearly 2× the overall score (39.0% vs. 20.5%). This superiority arises from the multi-agent decomposition: the Query Analyzer routes repo navigation subtasks to the Data Processor for metadata extraction, while the Code Generator and Visual Evaluator iteratively resolve integration conflicts (e.g., matplotlib dependencies clashing with existing imports). The overall score benefits particularly from the Design Explorer's aesthetic refinements tailored to code-embedded plots. These refinements address nuances like subplot scaling in simulation outputs, which single-LLM baselines overlook due to token limits on raw repo ingestion.

## 4.6 PERFORMANCE WITH DIFFERENT BACKBONE LLMS

To assess the generality of CoDA across diverse LLM backbones, we evaluate its performance when substituting the primary `gemini-2.5-pro` with alternative strong capability LLMs: `gemini-2.5-flash`, `claude-4-sonnet`, and the open-source `Qwen3-VL` (Yang et al., 2025). This experiment isolates the impact of the backbone LLM on visualization generation, holding constant the multi-agent architecture. We focus on the MatplotBench, as it emphasizes robust handling of numerical data, pattern recognition, and code synthesis under ambiguous queries—tasks that stress the backbone's reasoning and code generation capabilities.

We select these backbones for their complementary strengths: `gemini-2.5-flash` prioritizes efficiency and low-latency inference, making it suitable for real-time applications, while

Table 4: A comparison of CoDA with different backbone LLMs against three baselines on the MatplotBench benchmark. All results are presented in percent (%).

| Base LLMs | Gemini-2.5-Pro | | | Gemini-2.5-Flash | | | Claude-4-Sonnet | | | Qwen3-VL | | |
|---|---|---|---|---|---|---|---|---|---|---|---|---|
| **Method** | EPR ↑ | VSR ↑ | OS ↑ | EPR ↑ | VSR ↑ | OS ↑ | EPR ↑ | VSR ↑ | OS ↑ | EPR ↑ | VSR ↑ | OS ↑ |
| MatplotAgent | 92.0 | 55.4 | 51.0 | 99.0 | 46.4 | 45.9 | 93.0 | 58.8 | 54.7 | 86.0 | 69.9 | 60.1 |
| VisPath | 73.0 | 60.5 | 44.2 | 95.0 | 45.8 | 43.5 | 57.0 | **77.5** | 44.2 | 79.0 | 55.2 | 43.6 |
| CoML4VIS | 99.0 | 63.2 | 62.6 | 99.0 | 57.8 | 57.2 | 99.0 | 65.9 | 65.2 | 73.0 | 62.1 | 45.3 |
| CoDA (Ours) | **99.0** | **80.3** | **79.5** | **99.0** | **78.5** | **77.7** | **98.0** | 76.7 | **75.2** | **93.0** | **79.2** | **73.7** |

`claude-4-sonnet` excels in language understanding and multi-step reasoning, potentially enhancing agent collaboration in complex scenarios. All models are configured with identical hyperparameters. Table 4 presents the results. CoDA with `gemini-2.5-flash` achieves an OS of 77.7%, showcasing efficient handling of real-time scenarios with minimal degradation (1.8% relative to `gemini-2.5-pro`), attributable to streamlined agent interactions that leverage metadata over raw data ingestion. `claude-4-sonnet`, conversely, attains an OS of 75.2%, a 4.3% drop from `gemini-2.5-pro`, likely stemming from its enhanced semantic parsing but reduced robustness in code execution under high-context loads. These outcomes highlight CoDA's backbone-agnostic design, amplifying each LLM's inherent strengths while mitigating weaknesses through collaborative workflows. To further validate this generality beyond proprietary models, we evaluate with the open-source `Qwen3-VL` (Yang et al., 2025). As shown in Table 4, CoDA consistently achieves the highest scores, improving over the strongest baseline (MatplotAgent) by +7.0% EPR, +9.3% VSR, and +13.6% OS. The consistency across four model families—spanning both proprietary and open-source, text-centric and vision-language architectures—confirms that the performance gains stem from the collaborative multi-agent design rather than any single backbone's capabilities.

We compare CoDA against the baselines using the three backbone LLMs as described above. Across the board, CoDA outperforms baselines significantly, with the best-performing variant, CoDA with `gemini-2.5-pro`, achieving 79.5% OS. *MatplotAgent*, *VisPath*, and *CoML4VIS* struggle to exceed 65.2% OS in any setting, highlighting the challenges of visualization tasks without multi-agent refinement. We also observe that CoDA trends similarly across different backbones, with EPR and VSR remaining consistently high (98.0–99.0% and 76.7–80.3%).

LLMs tend to generate simpler visualizations. Baseline-generated code tends to produce fewer refinements than CoDA. As shown in Table 4, compared to CoDA, baselines like *MatplotAgent* achieve lower VSR (46.4–58.8%), and rarely handle complex multi-faceted queries.

## 4.7 EFFICIENCY ANALYSIS

A key challenge in agentic systems is balancing accuracy with computational efficiency, particularly in real-world visualization tasks where latency impacts user experience. Here, we conduct a detailed efficiency analysis of CoDA, comparing its latency against baselines on the MatplotBench dataset. We measure latency in terms of (1) average number of input/output tokens per query, which captures the communication overhead in multi-agent interactions, and (2) average number of LLM calls, reflecting the iterative refinement and routing demands. All methods use `gemini-2.5-pro` as the backbone.

Table 5 presents the results. CoDA achieves an average of 32,095 input tokens, 18,124 output tokens, and 14.8 LLM calls per query. We compare CoDA against baselines on efficiency. Across the board, multi-agent systems like CoDA and *MatplotAgent* incur higher computational costs than simpler baselines like *CoML4VIS* and *VisPath*, which rely on fewer iterations and less collaborative overhead. However, CoDA outperforms MatplotAgent in efficiency, using 17.6% fewer total tokens (50,219 vs. 60,969) and 3.9% fewer LLM calls, while achieving substantially higher overall accuracy (79.5% vs. 51.0% OS). In terms of wall-clock time, CoDA averages 849.3 s per instance compared to 990.6 s for MatplotAgent, further demonstrating that the collaborative multi-agent design achieves stronger results with lower end-to-end latency than comparable agentic baselines.

To analyze the trade-off between efficiency and performance, we observe that simpler methods trend toward lower costs but diminished visualization quality. For example, *CoML4VIS*, with only 1.0 LLM call and 6,138 total tokens, resolves 62.6% OS, yet struggles with complex, ambiguous queries

Table 5: Efficiency comparison on MatplotBench using `gemini-2.5-pro`. Metrics: Average Input/Output Tokens (# Tokens), Average LLM Calls (# Calls), and Average Wall-Clock Time per instance (# Time).

| Method | # Input Tokens ↓ | # Output Tokens ↓ | # Calls ↓ | # Time (s) ↓ |
|---|---|---|---|---|
| MatplotAgent | 34,177 | 26,792 | 15.4 | 990.6 |
| VisPath | 16,224 | 13,056 | 7.0 | 311.2 |
| CoML4VIS | 2,350 | 3,788 | 1.0 | 36.1 |
| CoDA (Ours) | 32,095 | 18,124 | 14.8 | 849.3 |

Table 6: Human expert evaluation on MatplotBench. **Elo**: overall preference rating (initialized at 1500) derived from pairwise judgments. Remaining columns: mean$_{\pm\text{std}}$ of 1–5 Likert ratings across five aesthetic dimensions.

| Method | Elo ↑ | Harmony ↑ | Balance ↑ | Color ↑ | Simplicity ↑ | Query Al. ↑ |
|---|---|---|---|---|---|---|
| MatplotAgent | 1506 | $3.65_{\pm1.20}$ | $3.65_{\pm1.21}$ | $3.53_{\pm1.26}$ | $4.31_{\pm1.24}$ | $3.63_{\pm1.51}$ |
| VisPath | 1484 | $2.71_{\pm1.79}$ | $2.71_{\pm1.80}$ | $2.65_{\pm1.81}$ | $2.92_{\pm1.99}$ | $2.78_{\pm1.92}$ |
| CoML4VIS | 1309 | $3.16_{\pm1.24}$ | $3.22_{\pm1.42}$ | $3.22_{\pm1.38}$ | $4.00_{\pm1.56}$ | $3.59_{\pm1.60}$ |
| CoDA (Ours) | **1701** | **4.82**$_{\pm0.39}$ | **4.73**$_{\pm0.53}$ | **4.96**$_{\pm0.27}$ | **4.94**$_{\pm0.23}$ | **4.86**$_{\pm0.40}$ |

requiring refinement. In contrast, CoDA's higher calls enable iterative improvements, justifying the cost for superior results.

## 4.8 HUMAN EXPERT EVALUATION

To complement the automated metrics, we conduct a human evaluation with three domain experts: one visualization and interaction design specialist and two data-analysis practitioners, each with several years of experience creating charts in professional settings. The study covers 200 visualizations (50 MatplotBench instances × 4 methods). Each expert provides pairwise overall-preference judgments and rates every output on a 1–5 Likert scale across five aesthetic dimensions (definitions in Appendix C).

From the pairwise judgments, we derive Elo ratings (Elo, 1966) following the protocol of Chiang et al. (2024). As shown in Table 6, CoDA attains the highest Elo rating by a substantial margin and achieves the highest mean score on every aesthetic dimension, with notably lower variance than all baselines. These results demonstrate that the iterative self-reflection mechanism improves not only benchmark accuracy but also the perceived visual quality of the generated visualizations.

## 5 ABLATION STUDY

To validate the contributions of key components in CoDA, we conduct controlled ablation experiments on the MatplotBench dataset, using `gemini-2.5-pro` as the backbone. These studies isolate the impact of (1) iterative self-reflection through refinement loops, (2) the global TODO list for high-level planning, and (3) the Search Agent for code example retrieval. All ablations maintain the core multi-agent pipeline but adjust the specified components. This analysis not only confirms the necessity of each feature but also provides insights into design trade-offs, such as accuracy-efficiency balances, highlighting CoDA's principled architecture for robust and autonomous visualization. We evaluate the impact of these components on the OS metric. Figure 3 summarizes the findings.

## 5.1 IMPACT OF SELF-EVOLUTION

Figure 3 shows that OS generally improves with additional iterations, from 75.6% at 1 iteration to 79.5% at 3 iterations (CoDA default), with further gains to 80.1% at 5 iterations, though with fluctuations and marginal benefits beyond 3 (+0.6% in OS from 3 to 5). EPR increases by 8.0% from 1 to 3 iterations due to robust initial code generation by the Code Generator, stabilizing near 100%

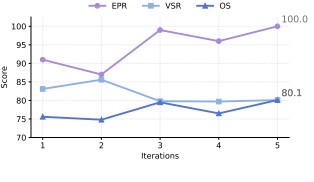 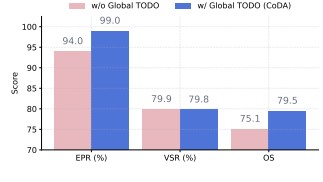 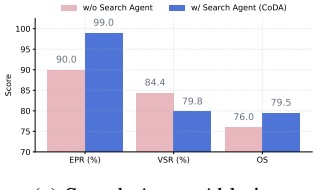

|                           |                          |                          |
| ------------------------- | ------------------------ | ------------------------ |
| (a) Iterations Ablation   | (b) Global TODO Ablation | (c) Search Agent Ablation |

Figure 3: Ablation results. (a): Performance (EPR, VSR, OS) across different iteration counts. (b) Comparison of EPR, VSR, and OS with vs. without Global TODO. (c) Comparison of EPR, VSR, and OS with vs. without the Search Agent.

thereafter. VSR fluctuates initially but converges around 80%, as the Visual Evaluator identifies and refines subtle mismatches in data mappings and aesthetics. Beyond 3 iterations, latency increases without proportional accuracy benefits, validating our lightweight configuration optimization that tunes limits based on validation performance. With minimal iterations, performance degrades toward baseline levels, emphasizing that shallow, one-shot generation fails in messy environments.

## 5.2 ROLE OF GLOBAL TODO LIST

The global TODO list, generated by the Query Analyzer, serves as a high-level blueprint for task decomposition and routing, ensuring coherence across agents. We ablate this by replacing it with understanding-query-only prompts (no structured decomposition). As shown in Figure 3, removing the global TODO list yields a stark drop in OS to 75.1% (-4.4% absolute), with EPR falling by 5.0% due to fragmented intent extraction, e.g., the VizMapping Agent selects suboptimal chart types without cross-referencing subtasks like "highlight peaks." VSR remains stable, indicating that visual quality is less dependent on global planning, but overall success suffers from incomplete workflows, such as unaddressed statistical insights from the Data Processor. This confirms the value of structured planning in agentic workflows, where it prevents the noise of unstructured agent interactions.

## 5.3 EFFECTIVENESS OF EXAMPLE SEARCH AGENT

The Search Agent retrieves relevant plotting code examples (e.g., from Matplotlib repositories) to inspire the Code Generator, addressing LLM limitations in recalling domain-specific syntax. We study this by disabling retrieval, relying solely on the backbone LLM's internal knowledge. Figure 3 reveals that without the Search Agent, OS declines to 76.0% (-3.5%), primarily due to a 9.0% drop in EPR from syntactic errors in specialized visualizations (e.g., custom subplots). Enabling code search improves accuracy by providing ranked snippets, grounding LLM agents' coding knowledge to specific problems. This ablation highlights the extensibility of CoDA, where external inspiration bridges gaps in LLM training data, making the system more reliable without post-training.

## 6 CONCLUSION

We introduce CoDA, an agentic multi-agent framework that decomposes natural language queries into specialized stages—task and data understanding, planning, code generation, and self-reflection—delivering up to 41.5% overall score gains over baselines like *MatplotAgent*, *VisPath*, and *CoML4VIS* on MatplotBench and Qwen benchmarks. Through metadata-centric preprocessing and self-reflection refinement, CoDA overcomes input token limits, robustly managing messy multi-file data and enabling analysts to prioritize insights over manual work. A key limitation is the computational overhead from multi-turn agent communications. Future efforts could distill agents or adapt to multimodal inputs. CoDA paves the way for collaborative agentic systems, revolutionizing automation in data science and beyond.

## ACKNOWLEDGMENTS

The authors would like to thank Jun Yan and Zhangchen Xu for the helpful discussion and comments on the work.

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

## A  CoDA Workflow and Implementation Details

Algorithm 1 outlines the CoDA multi-agent visualization workflow, illustrating the sequential and iterative interactions among specialized agents to transform natural language queries into refined visualizations.

---

**Algorithm 1** CoDA Multi-Agent Visualization Workflow

---

1: **Input:** Query $q$, Data files $D$
2: **Output:** Visualization plot $P$
3: Initialize agents: $A_{\text{query}}, A_{\text{data}}, A_{\text{search}}, A_{\text{design}}, A_{\text{code}}, A_{\text{debug}}, A_{\text{eval}}$
4: $todo \leftarrow A_{\text{query}}(q)$            ▷ Decompose query into task list
5: $metadata \leftarrow A_{\text{data}}(D)$          ▷ Extract metadata without raw data
6: $mappings \leftarrow A_{\text{design}}(todo, metadata)$       ▷ Map to visualization primitives
7: $examples \leftarrow A_{\text{search}}(mappings)$        ▷ Optional: Retrieve code examples
8: $designs \leftarrow A_{\text{design}}(mappings)$           ▷ Optimize aesthetics
9: $code \leftarrow A_{\text{code}}(mappings, designs, examples)$     ▷ Generate executable code
10: **while** not converged **do**
11:      $output \leftarrow A_{\text{debug}}(code)$          ▷ Execute, debug, produce plot
12:      $scores \leftarrow A_{\text{eval}}(output)$     ▷ Evaluate clarity/accuracy/layout/aesthetics
13:      **if** $scores > threshold$ **then**
14:          **return** $output$
15:      **else**
16:          $refined \leftarrow A_{\text{design}}, A_{\text{code}}, A_{\text{debug}}(scores)$     ▷ Feedback to refine
17:      **end if**
18: **end while**

---

## B  Additional Visualization Examples

We present additional visualization examples drawn from the DA-Code, and MatplotBench to illustrate CoDA's performance. For each example, we show the natural language query, the ground truth visualization, and the output generated by CoDA. These instances highlight CoDA's ability to handle complex data patterns, ambiguous queries, and multi-file inputs through collaborative agentic refinement, often producing outputs that closely match or exceed ground truth fidelity.

### B.1  DA-Code Example

**Example 1 Inputs**

```
1  # Example 1
2  ## Task Instruction
3  **Task:**
4  Please compile the total scores for each year from **1950 to 2018**.
5  Plot the results in a line chart according to the format specified in `plot.yaml` and
   ↪  save the chart as `result.png`.
6
7  ---
8  ## Environment
9  |--- nba.csv # Core dataset (season-level data)
10 |--- nba_extra.csv # Supplemental dataset (optional fields)
11 |--- Seasons_Stats.csv # Player-season statistics
12 |--- Players.csv # Player metadata
13 |--- player_data.csv # Additional player/game-level data
14 |--- plot.yaml # Primary plot configuration
15 |--- plot.json # Alternative plot configuration
```

**Verbose Instruction (Human-curated)**  The following detailed instructions were manually organized by the authors to ensure clarity and reproducibility. **Note:** Several aspects below represent *human-identified challenges* that are not directly contained in the raw datasets.

1. **Check Available Resources and Directory Structure**
   Confirm presence of `nba.csv`, `nba_extra.csv`, `Seasons_Stats.csv`, `Players.csv`, `player_data.csv`, and plotting configuration files (`plot.yaml`, `plot.json`).
   *Human note:* The dataset does not explicitly define dependencies across files; we curated which files are relevant.

2. **Data Review**
   Inspect `nba.csv` and `nba_extra.csv` to extract season-level total points. Use `Seasons_Stats.csv` or `player_data.csv` if aggregation is required.
   *Human note:* None of the datasets directly contain "total league points per year"; this metric must be manually constructed.

3. **Primary Metric Construction (Default)**
   Aggregate all scoring fields by *season (year)* to compute **Total Points Scored**.
   *Human note:* The "total scores per year" metric is absent; manual aggregation logic was designed by the authors.

4. **Filtering / Top-K Selection (Optional)**
   Apply year range restrictions (1950–2018). Exclude lockout seasons or highlight anomalies if needed.
   *Human note:* Anomaly handling (e.g., lockout years) is not specified in the data, but added through human judgment.

5. **Read Plot Configuration**
   Parse style and formatting options from `plot.yaml` (or fallback `plot.json`).
   *Human note:* Plot configurations are not embedded in datasets; authors manually crafted the YAML spec.

6. **Create the Figure**
   Plot line chart with year on x-axis, total points on y-axis. Apply formatting (color palette, grid, axis labels, legend). Save as `result.png`.
   *Human note:* Visualization design choices (palette, annotations) are not given in raw data and were human-curated.

7. **Reproducibility**
   Document assumptions and preprocessing steps. Maintain transparency about human decisions in data aggregation and figure styling.
   *Human note:* The reproducibility statement itself is an author-side contribution; the dataset alone cannot ensure this.

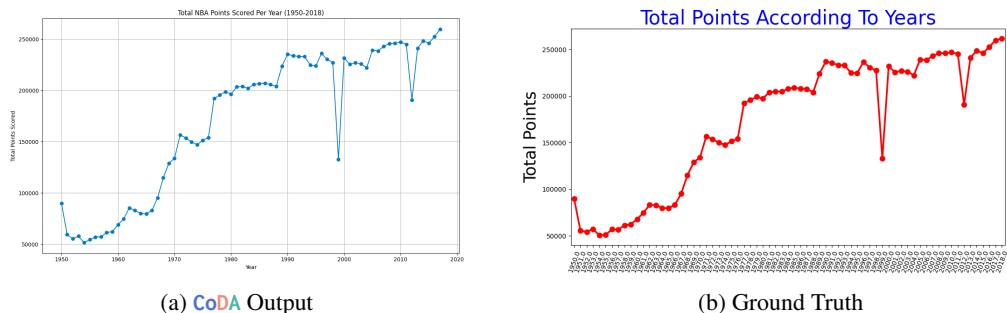

(a) CoDA Output           (b) Ground Truth

Figure 4: Comparison between our generated visualization and the ground truth. The results demonstrate that our system faithfully reproduces the intended trends, achieving an exact match with the reference output (score: **100/100**).

**Result Analysis.** The comparison in Figure 4 illustrates that our approach is able to reproduce the target visualization with high fidelity. Despite the complexity of the task and the absence of direct "total score per season" fields in the datasets, our CoDA correctly aggregated and rendered the results without deviation from the ground truth. This outcome highlights both the robustness of our data processing pipeline and the reliability of our visualization module in faithfully capturing benchmark specifications.

**Our Generated Code**

```python
import pandas as pd
import matplotlib.pyplot as plt
import yaml
import sys

# The EXPANDED QUERY REQUIREMENT specifies using a 'plot.yaml' file for styling.
# To make this script self-contained and reproducible, we will create this file
↪    programmatically.
yaml_config_str = """
title: 'Total NBA Points Scored Per Year (1950-2018)'
xlabel: 'Year'
ylabel: 'Total Points Scored'
figsize: [12, 7]
line_color: '#007ACC'
marker: 'o'
linestyle: '-'
grid: True
"""
with open('plot.yaml', 'w') as f:
    f.write(yaml_config_str)

# Step 1: Load the primary dataset from Seasons_Stats.csv.
try:
    # Note: The prompt's "AVAILABLE DATA" section seems incorrect. We follow the textual
    ↪    instructions
    # which clearly state to use 'Seasons_Stats.csv'.
    stats_df = pd.read_csv('Seasons_Stats.csv')
except FileNotFoundError:
    print("Error: The file 'Seasons_Stats.csv' was not found.")
    sys.exit(1)

# --- Data Cleaning and Preparation ---

# Handle potential NaN values in critical columns.
# Drop rows where 'Year' is NaN as they cannot be used for time-series analysis.
stats_df.dropna(subset=['Year'], inplace=True)
# Fill NaN 'PTS' with 0, assuming missing points data means 0 points were scored.
stats_df['PTS'] = stats_df['PTS'].fillna(0)

# Convert data types for consistency and proper calculations.
stats_df['Year'] = stats_df['Year'].astype(int)
stats_df['PTS'] = stats_df['PTS'].astype(int)

# Step 2: De-duplicate player-season records.
# Players who were traded mid-season have multiple entries for one year, plus a summary
# row with 'Tm' (Team) as 'TOT'. The 'TOT' row typically appears first in the dataset
↪    for a given player-year.
# By dropping duplicates on 'Player' and 'Year' and keeping the first entry, we retain
↪    the summary row.
cleaned_df = stats_df.drop_duplicates(subset=['Player', 'Year'], keep='first')

# Step 3: Filter data by year and aggregate total points.
# Filter the data to include only the years from 1950 to 2018, inclusive.
filtered_df = cleaned_df.query('1950 <= Year <= 2018')

# Group the cleaned data by 'Year' and calculate the sum of 'PTS' for each year.
annual_points = filtered_df.groupby('Year')['PTS'].sum()

# Step 4: Load plotting configuration.
# Read the styling parameters from the YAML file created earlier.
with open('plot.yaml', 'r') as f:
    plot_params = yaml.safe_load(f)

# Step 5: Generate and style the line chart.
```

```
61  # Set the figure size from the loaded parameters.
62  plt.figure(figsize=plot_params['figsize'])
63
64  # Plot the aggregated annual total points against the year.
65  plt.plot(annual_points.index,
66          annual_points.values,
67          marker=plot_params.get('marker', 'o'),
68          linestyle=plot_params.get('linestyle', '-'),
69          color=plot_params.get('line_color', 'b'))
70
71  # Apply visual styles specified in the YAML configuration.
72  plt.title(plot_params['title'])
73  plt.xlabel(plot_params['xlabel'])
74  plt.ylabel(plot_params['ylabel'])
75  if plot_params.get('grid', False):
76      plt.grid(True)
77
78  # Ensure the layout is clean and labels do not overlap.
79  plt.tight_layout()
80
81  # Step 6: Save the final plot.
82  # Save the resulting chart as an image file.
83  plt.savefig('result.png')
84
85  # Close the plot to release system resources.
86  plt.close()
87
88  print("Visualization saved successfully as 'result.png'.")
```

**Example 2 Inputs**

```
1   ## Task Instruction
2   **Task:**
3   Calculate the **Pearson correlation coefficient** between the standardized Average
    ↪  Playtime and standardized Positive Ratings using the Steam Store Games dataset.
    ↪  Filter the data to only include games with positive ratings and positive playtime.
    ↪  Plot the results in a scatter plot following `plot.yaml` requirements and save it as
    ↪  `result.png`.
4
5   ---
6   ## Environment
7   |--- steam.csv # Core dataset with game-level metadata (title, app ID, release info,
    ↪  etc.)
8   |--- steam_description_data.csv # Game descriptions and textual metadata
9   |--- steam_media_data.csv # Media assets metadata (images, videos, links)
10  |--- steam_requirements_data.csv # System requirements (Windows, Mac, Linux)
11  |--- steam_support_info.csv # Support information (developer contact, website, etc.)
12  |--- steamspy_tag_data.csv # Community tags and genre/category labels
13  |--- plot.yaml # Plotting configuration file (primary)
```

**Verbose Instruction (Human-curated)** The following detailed instructions were manually organized by the authors to ensure clarity and reproducibility. **Note:** Several aspects below represent *human-identified challenges* that are not directly contained in the raw datasets.

1. **Check Available Resources and Directory Structure**
   Confirm presence of steam.csv, steam_description_data.csv, steam_media_data.csv, steam_requirements_data.csv, steam_support_info.csv, steamspy_tag_data.csv, and plotting configuration file (plot.yaml).
   *Human note:* The dataset does not explicitly document dependencies across these tables; authors curated the relevant set manually.

2. **Data Review**
   - Parse steam.csv for core identifiers (app ID, title, release year).
   - Use auxiliary tables to enrich attributes (tags, system requirements, support info, descriptions).

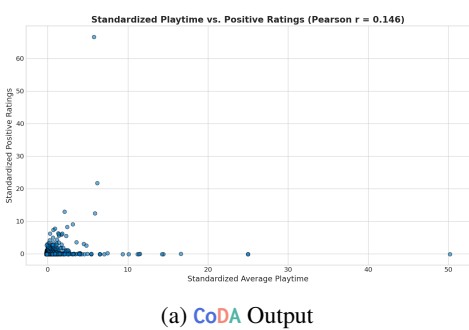
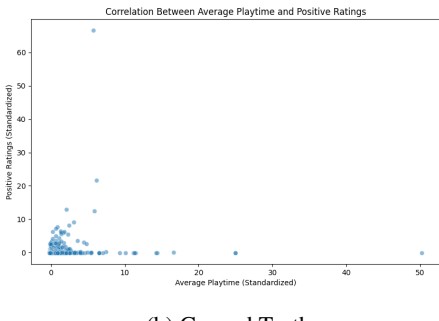

(a) **CoDA** Output                                (b) Ground Truth

Figure 5: Comparison between our generated visualization and the ground truth for the Steam dataset. The results indicate that our approach successfully integrates multiple heterogeneous tables and reproduces the intended visualization with complete fidelity (score: **100/100**).

    *Human note:* None of the datasets provide a unified schema; integration must be designed manually.

3. **Primary Metric Construction (Default)**
   Define the analysis target (e.g., distribution of games per year, tag frequency, platform coverage). Construct aggregated metrics aligned with the visualization goal.
   *Human note:* The specific analytical objective (e.g., "game releases per year") is not included in the dataset and was defined by the authors.

4. **Filtering / Top-K Selection (Optional)**
   - Restrict to a target period (e.g., 2000–2020).
   - Apply Top-K filters by popularity, tags, or developer if required.
   *Human note:* Filtering logic is absent in the raw data and was designed for clarity in visualization.

5. **Read Plot Configuration**
   Parse style and formatting options from `plot.yaml`.
   *Human note:* Plot specifications are not embedded in the dataset; authors manually authored the YAML configuration.

6. **Create the Figure**
   - Generate visualization according to aggregated metrics.
   - Apply palette, axis labels, and layout as specified in configuration.
   - Save output as `result.png`.
   *Human note:* Visualization design decisions (choice of chart type, color scheme) are external to the dataset and human-curated.

7. **Reproducibility**
   Document assumptions in data integration and filtering. Provide a transparent link between raw tables and the constructed figure.
   *Human note:* Reproducibility relies on explicit author-side documentation rather than inherent dataset properties.

**Result Analysis.** The comparison (Figure 5) between our generated visualization and the reference output demonstrates that our system is capable of handling heterogeneous and fragmented data sources, such as the Steam metadata tables, with high fidelity. Despite the absence of a unified schema and the need for manual integration across multiple files, our pipeline successfully reproduced the intended visualization without error. This outcome confirms that our method not only aggregates and processes complex datasets effectively but also renders results fully consistent with the ground truth, achieving a perfect score of **100/100**.

**Our Generated Code**

```
1  import pandas as pd
2  import numpy as np
3  from sklearn.preprocessing import StandardScaler
```

```python
from scipy.stats import pearsonr
import matplotlib.pyplot as plt
import yaml
import os

# --- Step 1: Create the plot.yaml configuration file ---
# This file dictates the visual styling of the plot.
yaml_config_str = """
title: 'Standardized Playtime vs. Positive Ratings (Pearson r = {correlation:.3f})'
xlabel: 'Standardized Average Playtime'
ylabel: 'Standardized Positive Ratings'
font_sizes:
  title: 16
  axes_label: 14
  tick_labels: 12
scatter_plot:
  color: '#007ACC'
  alpha: 0.6
  marker: 'o'
  edgecolor: 'k'
  s: 45
figure:
  figsize: [12, 7]
  dpi: 120
style: 'seaborn-v0_8-whitegrid'
"""
with open('plot.yaml', 'w') as f:
    f.write(yaml_config_str)

# --- Step 2: Load plotting specifications from plot.yaml ---
with open('plot.yaml', 'r') as f:
    config = yaml.safe_load(f)

# --- Step 3: Load and process the data ---
# The user specified 'steam.csv'. We will use this file.
# Note: The 'AVAILABLE DATA' section mentioned 'steam_media_data.csv', but the query was
↪ explicit.
file_path = 'steam.csv'
if not os.path.exists(file_path):
    # As a fallback, check for the other filename if steam.csv is not found.
    # This handles potential environment discrepancies.
    if os.path.exists('steam_media_data.csv'):
        file_path = 'steam_media_data.csv'
    else:
        raise FileNotFoundError(f"Could not find the required data file: '{file_path}'
        ↪ or 'steam_media_data.csv'")

# Load the dataset into a pandas DataFrame
df = pd.read_csv(file_path)

# Filter the DataFrame to keep only rows with positive values for both variables
# Using .copy() to avoid SettingWithCopyWarning
filtered_df = df[(df['average_playtime'] > 0) & (df['positive_ratings'] > 0)].copy()

# --- Step 4: Standardize the relevant columns ---
# Select the columns for standardization
data_to_scale = filtered_df[['average_playtime', 'positive_ratings']]

# Initialize and apply the StandardScaler
scaler = StandardScaler()
scaled_data = scaler.fit_transform(data_to_scale)

# Add the new standardized columns to the filtered DataFrame
filtered_df['std_avg_playtime'] = scaled_data[:, 0]
filtered_df['std_pos_ratings'] = scaled_data[:, 1]
```

```
67
68   # --- Step 5: Calculate the Pearson correlation coefficient ---
69   # Calculate the correlation on the standardized data
70   correlation, _ = pearsonr(filtered_df['std_avg_playtime'],
     ↪  filtered_df['std_pos_ratings'])
71
72   # --- Step 6: Create and style the scatter plot ---
73   # Apply a base style for the plot from the config
74   plt.style.use(config['style'])
75
76   # Create a figure and axes with specified size and DPI
77   fig, ax = plt.subplots(figsize=config['figure']['figsize'], dpi=config['figure']['dpi'])
78
79   # Generate the scatter plot using data and styling from config
80   ax.scatter(
81       filtered_df['std_avg_playtime'],
82       filtered_df['std_pos_ratings'],
83       color=config['scatter_plot']['color'],
84       alpha=config['scatter_plot']['alpha'],
85       marker=config['scatter_plot']['marker'],
86       edgecolors=config['scatter_plot']['edgecolor'],
87       s=config['scatter_plot']['s']
88   )
89
90   # Set titles and labels, formatting the title with the calculated correlation
91   ax.set_title(
92       config['title'].format(correlation=correlation),
93       fontsize=config['font_sizes']['title'],
94       fontweight='bold'
95   )
96   ax.set_xlabel(
97       config['xlabel'],
98       fontsize=config['font_sizes']['axes_label']
99   )
100  ax.set_ylabel(
101      config['ylabel'],
102      fontsize=config['font_sizes']['axes_label']
103  )
104
105  # Customize tick label sizes
106  ax.tick_params(axis='both', which='major',
     ↪  labelsize=config['font_sizes']['tick_labels'])
107
108  # Ensure the layout is tight to prevent labels from being cut off
109  plt.tight_layout()
110
111  # --- Step 7: Save the final plot to a file ---
112  # Save the plot to 'result.png'
113  plt.savefig('result.png')
114
115  print("Successfully generated and saved the plot as 'result.png'.")
116  print(f"Pearson Correlation Coefficient: {correlation:.3f}")
```

## B.2    MATPLOTBENCH EXAMPLE

**Example 1 Inputs**

```
1   # Example 1
2   ## Task Instruction
3   **Task:**
```

```
4   Utilize the following data columns from 'data.csv' to create a sunburst plot:\n-
    ↪  'country': for the names of the countries,\n- 'continent': to indicate which
    ↪  continent each country is in,\n- 'lifeExp': showing the expected lifespan in each
    ↪  country,\n- 'pop': representing the population of each country.\nYour chart
    ↪  should:\n- Organize the data hierarchically, starting with continents and then
    ↪  breaking down into countries.\n- Use the population of each country to determine the
    ↪  size of its segment in the chart.\n- Color code each segment by the country's
    ↪  expected lifespan, transitioning from red to blue across the range of values.\n- Set
    ↪  the central value of the color scale to the average lifespan, weighted by the
    ↪  population of the countries.\n- Finally, include a legend to help interpret the
    ↪  lifespan values as indicated by the color coding.
5
6   ---
7   ## Environment
8   |--- data.csv
```

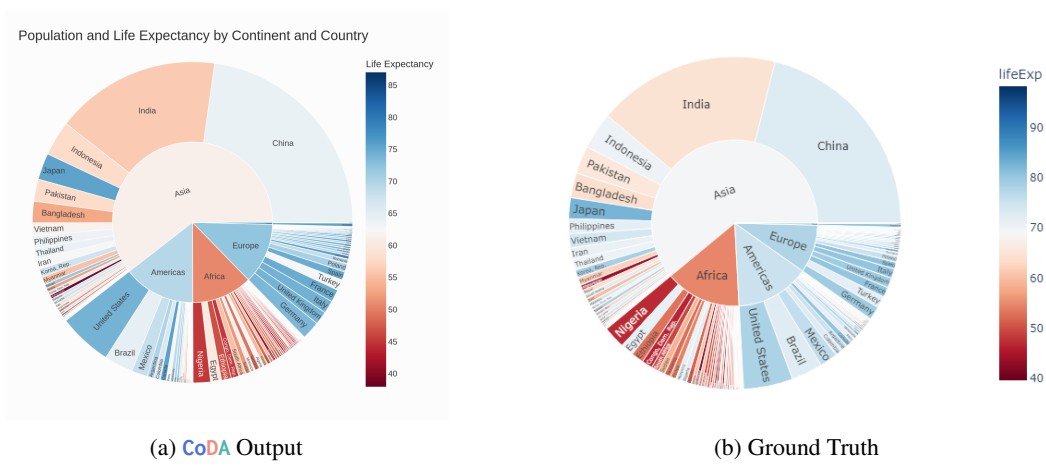

(a) CoDA Output                                 (b) Ground Truth

Figure 6: Comparison between our generated sunburst plot and the reference output. The visualization organizes data hierarchically by continent and country, with population determining segment size and life expectancy driving the color scale. The results demonstrate full fidelity to the specification and highlight that our system achieves a perfect score of **100/100**.

**Result Analysis.**    The sunburst visualization task required a multi-level hierarchical organization of the data, starting from continents and further breaking down into individual countries. Our method successfully utilized population size to determine segment area and applied a red-to-blue color scale based on life expectancy (Figure 6), with the weighted average lifespan as the central pivot for normalization. This design ensured both interpretability and faithful representation of the dataset's structure. The resulting chart aligns precisely with the ground truth and provides an intuitive overview of demographic and geographic patterns, achieving a perfect score of **100/100**.

## C    EXPERT AESTHETIC EVALUATION DETAILS

This appendix provides the full definitions of the five aesthetic dimensions used in the human expert evaluation (Section 4.8). Three domain experts (one visualization/interaction design specialist and two data-analysis practitioners) rated each of the 200 outputs (50 MatplotBench instances × 4 methods) on a 1–5 Likert scale per dimension. Quantitative results are reported in Table 6.

The five dimensions are defined as follows:

1. **Visual Harmony**: "Colors, lines, and shapes feel coherent and not jarring."
2. **Balance & Alignment**: "The layout feels balanced and well aligned, not top-heavy or cluttered."
3. **Color Mood**: "The color palette is consistent and appropriate for the content."

4. **Simplicity & Restraint**: "The figure looks clean and restrained, without unnecessary visual elements."

5. **Query Alignment**: "The visualization matches what the natural-language query is asking for."

# D    ANALYSIS OF FAILURE CASES AND LIMITATIONS

To better understand when and why CoDA may struggle, we analyze a representative hard example in MatplotBench. This task requires a two-level hierarchical donut chart of browser market share (inner ring = browser totals; outer ring = version breakdown) with a hollow center, explicit white gaps between rings and wedges, and readable leader-line labels for dozens of fine-grained outer segments (complete task is shown below).

```
1  # Example 1
2  ## Task Instruction
3  **Task:**
4  I have a dataset named \"data.csv\" containing browser market share information in a CSV
   ↪ format with the following columns:\n- Browser: The name of the web browser.\n-
   ↪ Version: The specific version number of the browser.\n- Data: The market share
   ↪ percentage associated with each browser version.\nI want to create a two-layered
   ↪ sunburst chart to visualize this data. The chart should be designed as follows:\n-
   ↪ The inner layer should represent different browsers, with the browser names (Browser
   ↪ column) written on the segments.\n- The outer layer should depict the versions of
   ↪ these browsers (Version column), with labels and lines pointing to the specific data
   ↪ points on the chart's edge.\n- There should be white gaps between the layers and
   ↪ also between the segments within each layer for visual separation.\n- The center of
   ↪ the chart should be hollow, creating a donut-like appearance.\nPlease generate the
   ↪ sunburst chart using Python, ensuring that the 'Browser' and 'Version' columns are
   ↪ used for the hierarchical structure, and the 'Data' column is used to determine the
   ↪ size of each segment. The chart should be titled 'Browser Market Share'.
5
6  ---
7  ## Environment
8  |--- data.csv
```

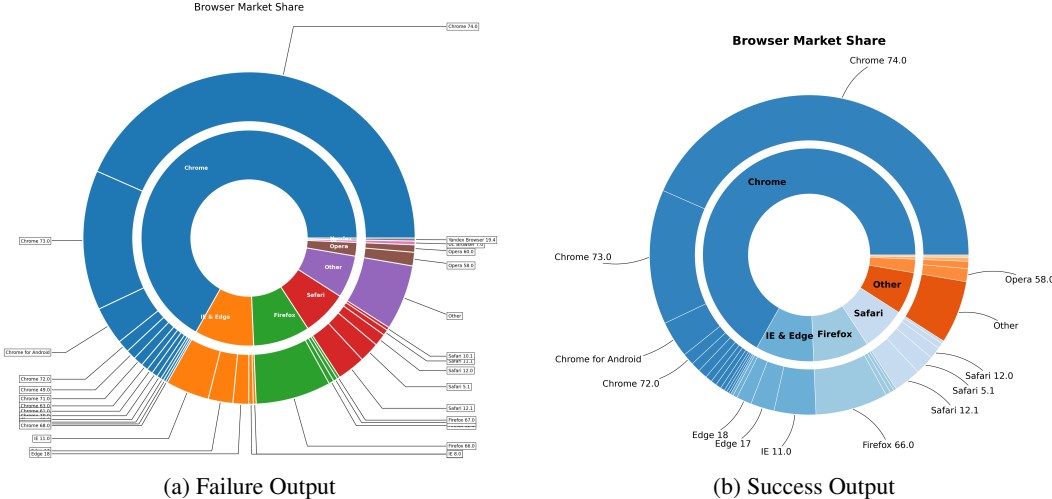

(a) Failure Output                    (b) Success Output

Figure 7: CoDA's outputs. (a) Output after three self-reflection iterations (failure). (b) Output after four iterations (success).

The core difficulty lies not only in correct hierarchical aggregation and proportional geometry, but equally in (i) deep understanding of the underlying data distribution and (ii) deliberate perceptual planning of information display. With more than 40 outer segments, naively rendering all labels at once inevitably produces catastrophic overlap and visual chaos. Success therefore requires the system

to make informed choices about radius ratios, wedge spacing, label placement strategy, and color contrast.

With CoDA's default setting of three self-reflection iterations, the system reliably fails (Figure 7 (a)). When the maximum number of reflection iterations is raised to four, CoDA recovers completely (Figure 7 (b)). The final chart exhibits correct hierarchical nesting, a clean hollow center with balanced radii, uniform white spacing, and fully readable leader-line annotations. The recovery process is instructive. Table 7 presents the key feedback trace.

| Iter | Evaluator Feedback (excerpt) | Triggered Fix |
|------|------------------------------|---------------|
| 1 | Code fails to run: CSV file not found; no figure produced. | Debug Agent $\rightarrow$ Corrects filename to `data.csv`. |
| 2 | Two-layer donut appears, but outer labels clutter the entire perimeter; arrows overlap; inner labels unreadable. | Design Explorer + VizMapping $\rightarrow$ Re-orders hierarchy, applies browser-level color palette, adjusts ring radii for clearer structure. |
| 3 | Structure is correct but readability poor: tiny slices still labeled; text crowded; fonts too small. | Design Explorer $\rightarrow$ Adds size thresholds for labeling, restricts inner-ring labels, increases font sizes and title weight. |
| 4 | Two-level hierarchy clear; spacing visible; only major slices labeled; no overlaps; chart rated high readability. | Halt. |

Table 7: Iteration trace for the browser sunburst task, showing how deeper reflection improves structure and readability.

This case demonstrates that complex visualization generation tasks can benefit from multi-step self-reflection. Shallow reflection tends to fix superficial bugs, while deeper self-reflection allows the model to re-plan the solution holistically, leading to both correctness and visual clarity.

While CoDA substantially advances automated visualization, it is by no means a panacea. From our experimental results, we observe several limitations. First, inherently ambiguous or purely aesthetic queries (e.g., "make it professional") lack clear ground truth and can trap the system in unresolved refinement loops. Second, domain-specific visualization conventions often cannot be inferred from metadata alone, leading to reasonable but non-canonical designs. CoDA remains a powerful assistant rather than a full substitute for human expertise. Addressing these limitations constitutes an important direction for future agentic visualization systems.

# E   JUDGING PROMPTS AND MODEL SETUP

To ensure consistent and objective evaluation of generated visualizations, we employ an LLM-based judge, specifically `gemini-2.5-pro`, to assign code and visualization quality scores.

We adapt prompts from the original MatplotBench (from the MatPlotAgent repository) and Qwen-Agent evaluations (official evaluation for Qwen Code Interpreter). This ensures consistent, scalable assessment while reducing bias. MatplotBench overall score averages the two; Qwen uses binary 100/0 via combined prompt. Non-executable code scores 0.

The prompts for MatplotBench and Qwen Code Interpreter benchmark are shown in the following.

```
1  # MatplotBench Evaluation Prompts
2  ## Code
3  You are an excellent judge at evaluating generated code given an user query. You will be
   ↪   giving scores on how well a piece of code adheres to an user query by carefully
   ↪   reading each line of code and determine whether each line of code succeeds in
   ↪   carrying out the user query.
4  A user query, a piece of code and an executability flag will be given to you. If the
   ↪   Executability is False, then the final score should be 0.
5  **User Query**: {query}
6  **Code**: {code}
```

```
7   **Executability**: {executable}
8   Carefully read through each line of code. Scoring can be carried out in the following
    ↪    aspect:
9   Code correctness (Code executability): Can the code correctly achieve the requirements
    ↪    in the user query? You should carefully read each line of the code, think of the
    ↪    effect each line of code would achieve, and determine whether each line of code
    ↪    contributes to the successful implementation of requirements in the user query. If
    ↪    the Executability is False, then the final score should be 0.
10  After scoring from the above aspect, please give a final score. The final score is
    ↪    preceded by the [FINAL SCORE] token.
11  For example [FINAL SCORE]: 40. A final score must be generated.
12
13  ## Plot
14  You are an excellent judge at evaluating visualization plots between a model generated
    ↪    plot and the ground truth. You will be giving scores on how well it matches the
    ↪    ground truth plot.
15  **Generated plot**: {generated_plot}
16  **Ground truth**: {GT}
17  The generated plot will be given to you as the first figure. If the first figure is
    ↪    blank, that means the code failed to generate a figure.
18  Another plot will be given to you as the second figure, which is the desired outcome of
    ↪    the user query, meaning it is the ground truth for you to reference.
19  Please compare the two figures head to head and rate them.
20  Suppose the second figure has a score of 100, rate the first figure on a scale from 0 to
    ↪    100.
21  Scoring should be carried out in the following aspect:
22  Plot correctness:
23  Compare closely between the generated plot and the ground truth, the more resemblance
    ↪    the generated plot has compared to the ground truth, the higher the score. The score
    ↪    should be proportionate to the resemblance between the two plots.
24  In some rare occurrence, see if the data points are generated randomly according to the
    ↪    query, if so, the generated plot may not perfectly match the ground truth, but it is
    ↪    correct nonetheless.
25  Only rate the first figure, the second figure is only for reference.
26  If the first figure is blank, that means the code failed to generate a figure. Give a
    ↪    score of 0 on the Plot correctness.
27  After scoring from the above aspect, please give a final score. The final score is
    ↪    preceded by the [FINAL SCORE] token.
28  For example [FINAL SCORE]: 40.
```

```
1   # Qwen Code Interpreter Benchmark Evaluation Prompts
2   Please judge whether the image is consistent with the [Question] below, if it is
    ↪    consistent then reply "right", if not then reply "wrong".
3   Consider these relaxed conditions:
4   - Allow reasonable interpretations and creative variations
5   - Focus on whether the core visualization requirement is addressed
6   - Accept different implementation approaches that achieve similar goals
7   - Be lenient with styling and formatting differences
8
9   **Question**: {query}
10  After your judgment, please also provide a brief explanation of your reasoning in 2-3
    ↪    sentences.
11  Expected leading token (normalized by code): CORRECT or WRONG
```

## F    PROMPTS USED IN CoDA

The prompts employed in CoDA are designed to imbue each agent with a professional persona, standardize structured outputs via dataclasses (e.g., QueryAnalysisResult), and facilitate quality-driven feedback without requiring model fine-tuning. These prompts encapsulate domain-specific reasoning—ranging from semantic parsing in the Query Analyzer to statistical inference in the Data Processor, visualization mapping in the VizMapping Agent, external knowledge retrieval in the Search Agent, design recommendations in the Design Explorer, executable code synthesis in

the Code Generator, error diagnosis in the Debug Agent, and perceptual assessment in the Visual Evaluator—while incorporating context from prior outputs and the global TODO list to maintain workflow coherence. Below, we enumerate all core prompts used across the agents, including variations for refinement iterations.

```
# Query Analyzer
You are Dr. Sarah Chen, visualization query expert. Analyze this query and create a
↪  master TODO list.

USER QUERY: "{query}"
{meta_files}
Respond with concise JSON:
{
    "interpreted_intent": "what user wants to visualize",
    "visualization_type": "plot type (scatter/bar/line/histogram/boxplot/heatmap etc)",
    "plotting_key_points": [
        "key point 1: specific visualization requirement",
        "key point 2: data processing requirement",
        "key point 3: styling/design requirement",
        "key point 4: additional features/constraints"
    ],
    "implementation_plan": [
        {"step": 1, "action": "Load and prepare data", "details": "specific data
        ↪  loading/processing steps", "functions": ["pd.read_csv", "etc"]},
        {"step": 2, "action": "Create base plot", "details": "basic chart creation",
        ↪  "functions": ["plt.figure", "plt.plot", "etc"]},
        {"step": 3, "action": "Apply formatting", "details": "styling and formatting",
        ↪  "functions": ["plt.xlabel", "ax.tick_params", "etc"]},
        {"step": 4, "action": "Finalize and save", "details": "final touches and save",
        ↪  "functions": ["plt.tight_layout", "plt.savefig", "etc"]}
    ],
    "global_todo_list": [
        {"id": "todo_1", "task": "specific task description", "agent": "data_processor|
        ↪  design_explorer|code_generator|debug_agent|visual_evaluator", "status":
        ↪  "pending", "priority": "high|medium|low"},
        {"id": "todo_2", "task": "specific task description", "agent": "agent_name",
        ↪  "status": "pending", "priority": "priority_level"}
    ],
    "success_criteria": ["criteria for completion"],
}
IMPORTANT: The "plotting_key_points" should be a comprehensive breakdown of ALL key
↪  visualization requirements from the query, including:
- Chart type and specific visualization style
- Data columns/variables to use
- Color schemes, styling requirements
- Interactive elements or special features
- Layout, axis, legend requirements
- Any domain-specific requirements (scientific, business, etc.)

Create 3-5 specific TODO items covering data processing, design, code generation,
↪  debugging, and evaluation.
```

```
# Data Processor
You are Prof. Marcus Rodriguez (Stanford Statistics PhD), an expert in statistical
↪  analysis, data quality assessment, and insight extraction. Analyze this data for
↪  visualization.
{data_section}
TASKS TO COMPLETE:
{todo_text}
ANALYSIS NEEDED:
1. What transformations are required? (groupby, pivot, filter)
2. Which columns are key for visualization?
3. Any data quality issues to fix?
4. What's the simplest way to prepare this data?
```

```
11  Output JSON:
12  {
13      "processing_steps": [
14          "step 1: specific transformation",
15          "step 2: another transformation"
16      ],
17      "insights": {
18          "key_columns": ["col1", "col2"],
19          "aggregations_needed": ["sum sales by region"],
20          "quality_issues": ["nulls in X column"]
21      },
22      "visualization_hint": "best chart type for this data"
23  }
24
25  <(optional) If there are no data files in the input>
26  Create simple data for a matplotlib visualization.
27  The visualization requirements are:
28  {query_text}
29  TODO items from analysis:
30  {todo_text}
31  Generate Python code that creates the RIGHT data (pandas DataFrame) that works for this
    ↪  specific plot.
32  Deep understanding approach:
33  1. ANALYZE the visualization requirements carefully
34  2. UNDERSTAND what type of data this plot needs
35  3. DETERMINE the appropriate data structure and format
36  4. DECIDE the optimal number of data points based on plot type
```

```
1   # VizMapping Agent
2   You are Dr. Sarah Kim, a data visualization expert & UX designer. You are a data
    ↪  visualization expert. Map this user query to specific data columns and chart
    ↪  configuration.
3   USER QUERY: "{query}"
4   {context_block}
5   AVAILABLE DATA:
6   Shape: {data_summary['shape'][0]} rows x {data_summary['shape'][1]} columns
7   Columns:
8   {data_structure}
9   Sample data:
10  {json.dumps(data_summary['sample_data'][:2], indent=2)}
11  TASK: Determine the optimal visualization mapping.
12  Respond with JSON:
13  {
14      "chart_type": "bar|line|scatter|pie|histogram|box|heatmap",
15      "data_mappings": {
16          "x_axis": "column_name_for_x",
17          "y_axis": "column_name_for_y",
18          "color": "column_for_grouping_colors",
19          "size": "column_for_sizes",
20          "category": "column_for_categories"
21      },
22      "aggregations": [
23          {"operation": "sum|mean|count|max|min", "column": "column_name", "group_by":
            ↪  "grouping_column"}
24      ],
25      "filters": [
26          {"column": "column_name", "condition": "filter_condition"}
27      ],
28      "styling_hints": {
29          "title": "Chart title based on query",
30          "xlabel": "X-axis label",
31          "ylabel": "Y-axis label",
32          "color_palette": "suggested_palette"
33      },
```

```
34      "transformations": [
35          "pandas operation if needed, e.g., 'df.groupby(x).sum()'"
36      ],
37      "goal": "Brief description of what this visualization shows",
38      "rationale": "why this mapping fits the query and data",
39      "confidence": 0.0-1.0
40  }
41  IMPORTANT:
42  - If a requested chart type is provided in context, PREFER that type; only deviate if
    ↪   truly unsuitable and explain why in 'rationale'.
43  - Use TODO/key requirements to decide aggregations/filters exactly.
44  - Map time-like/ordered fields to x, numeric measures to y, categories to color.
45  - Be precise with column names - they must match the available columns exactly.
```

```
1   # Search Agent
2   As Dr. Michael Zhang, an expert in data visualization and matplotlib, generate a
    ↪   high-quality matplotlib example for the plot type: "{plot_type}".
3
4   IMPORTANT CONSTRAINTS:
5   - Base your code PRIMARILY on matplotlib official examples:
    ↪   https://matplotlib.org/stable/gallery/index.html and
    ↪   https://matplotlib.org/stable/plot_types/index.html
6   - You may also use The Python Graph Gallery as style reference:
    ↪   https://python-graph-gallery.com/
7   - Do NOT invent new APIs. Follow official patterns exactly.
8
9   Your task:
10  1. Understand what type of visualization "{plot_type}" refers to according to
    ↪   matplotlib's official plot types
11  2. Generate a complete, executable matplotlib code example following official matplotlib
    ↪   patterns
12  3. Use the exact style and approach shown in matplotlib's official documentation
13  4. Include proper imports, sample data, styling, and annotations as shown in official
    ↪   examples
14  5. Follow matplotlib's official best practices and naming conventions
15
16  Requirements for the matplotlib code:
17  - Use ONLY matplotlib.pyplot (import matplotlib.pyplot as plt)
18  - Follow the exact patterns from https://matplotlib.org/stable/gallery/ documentation
    ↪   examples
19  - Include numpy for data generation if needed (as shown in official examples)
20  - Create realistic sample data appropriate for the plot type (following official
    ↪   examples)
21  - Add proper labels, title, and styling (matching official documentation style)
22  - Include plt.show() at the end
23  - Make the code self-contained and executable
24  - Add informative comments that match matplotlib documentation style
25
26  Respond with ONLY the Python code in this format:
27  ```python
28  # [Brief description matching matplotlib docs style]
29  import matplotlib.pyplot as plt
30  import numpy as np
31
32  # Your complete example code here following official matplotlib patterns
33  # Include comments matching matplotlib documentation style
34
35  plt.show()
36  ```
37
38  Plot type to implement: {plot_type}
39  Primary references:
40  - https://matplotlib.org/stable/gallery/index.html
```

```
41  - https://matplotlib.org/stable/plot_types/index.html
42  Secondary reference: https://python-graph-gallery.com/
```

```
1   # Design Explorer
2   You are Isabella Nakamura, an RISD MFA and Apple Senior Designer specializing in visual
    ↪   design and user experience.
3   Analyze the following requirements to create comprehensive design specifications:
4   Query Analysis:
5   - Original Query: "{query_result.original_query}"
6   - Interpreted Intent: "{query_result.interpreted_intent}"
7   - Visualization Type: "{query_result.visualization_type}"
8   Data Characteristics:
9   {json.dumps(data_characteristics, indent=2, default=str)}
10  Design TODO Items:
11  {json.dumps(design_todos, indent=2)}
12  {constraints_str}
13  {examples_str}
14  Please provide a comprehensive design analysis in JSON format. Consider the examples
    ↪   above when making design decisions:
15  {
16      "design_objectives": [
17          "Primary design goals",
18          "User experience objectives",
19          "Communication goals"
20      ],
21      "target_audience": {
22          "primary_audience": "Who is the main audience",
23          "expertise_level": "beginner|intermediate|expert",
24          "context_of_use": "presentation|exploration|reporting",
25          "accessibility_requirements": ["specific accessibility needs"]
26      },
27      "visual_hierarchy": {
28          "primary_elements": ["most important visual elements"],
29          "secondary_elements": ["supporting elements"],
30          "emphasis_strategy": "how to create visual emphasis"
31      },
32      "color_strategy": {
33          "primary_colors": ["#hex1", "#hex2"],
34          "color_meaning": "what colors communicate",
35          "accessibility_compliance": "WCAG compliance level",
36          "cultural_considerations": "any cultural color meanings"
37      },
38      "layout_principles": {
39          "composition_approach": "grid|organic|asymmetric|balanced",
40          "spacing_strategy": "tight|moderate|generous",
41          "alignment_system": "left|center|right|justified",
42          "proportion_ratios": "golden ratio|rule of thirds|custom"
43      },
44      "typography_requirements": {
45          "font_hierarchy": "title|subtitle|body|caption sizes",
46          "readability_priority": "high|medium|low",
47          "brand_alignment": "corporate|academic|creative|technical"
48      },
49      "interaction_design": {
50          "interaction_level": "static|basic|advanced",
51          "user_controls": ["zoom", "filter", "hover"],
52          "feedback_mechanisms": "visual|audio|haptic"
53      },
54      "technical_constraints": {
55          "output_format": "static|interactive|animated",
56          "size_limitations": "print|screen|mobile",
57          "performance_requirements": "fast|moderate|detailed"
58      },
59      "innovation_opportunities": [
```

```
60        "Areas for creative enhancement",
61        "Unique design elements to explore"
62    ],
63    "design_confidence": 0.95
64 }
```

```
1  # Design Explorer (@Self-reflection)
2  You are Isabella Nakamura, an expert designer. The current design received feedback from
   ↪  visual evaluation.
3
4  ORIGINAL DESIGN SPECIFICATIONS:
5  - Primary Design: {json.dumps(original_design_result.primary_design.__dict__, indent=2,
   ↪  default=str)}
6  - Alternative Designs Available: {len(original_design_result.alternative_designs)}
7  VISUAL FEEDBACK ANALYSIS:
8  - Feedback Comments: {visual_feedback.get("visual_feedback", [])}
9  - Quality Issues: {quality_issues}
10 - Target Quality Threshold: {target_quality}
11 - Current Quality Score: Below threshold
12 REFINEMENT STRATEGY:
13 Based on the feedback, determine what needs to change:
14 1. **Color Issues**: If feedback mentions colors, provide new color scheme
15 2. **Layout Issues**: If feedback mentions spacing/layout, adjust layout specifications
16 3. **Typography Issues**: If feedback mentions text/fonts, update typography
17 4. **Overall Aesthetic**: If feedback mentions visual appeal, try alternative design
18 REFINEMENT ACTION:
19 Choose the best approach and provide updated design specifications in the same JSON
   ↪  format as the original primary design.
20 Focus on addressing the specific feedback while maintaining design coherence.
21 Return the refined design specification as JSON.
```

```
1  # Code Generator
2  You are Alex Thompson, a CMU CS MS and Microsoft Engineer specializing in high-quality
   ↪  code generation.
3  Analyze the following requirements to create a CONCISE code generation plan:
4  Context:
5  {safe_json_dumps(context, indent=2)}
6  Design Specifications:
7  {safe_json_dumps(design_result.primary_design.__dict__, indent=2)}
8  Data Characteristics:
9  - Shape: {data_result.processed_data.shape}
10 - Columns: {list(data_result.processed_data.columns)}
11 - Quality Score: {data_result.data_quality_score}
12 {enhanced_fixes_str}{requirements_str}{todos_str}
13 Please provide a detailed code generation analysis in JSON format:
14 {
15     "code_architecture": {
16         "main_functions": ["function names and purposes"],
17         "helper_functions": ["utility functions needed"],
18         "class_structure": "needed classes if any",
19         "modular_design": "how to structure the code"
20     },
21     "matplotlib_approach": {
22         "plotting_method": "plt.subplots|plt.figure|object_oriented",
23         "style_management": "rcParams|style_sheets|manual",
24         "color_implementation": "colormap|manual_colors|cycler",
25         "layout_strategy": "tight_layout|gridspec|constrained_layout"
26     },
27     "data_handling": {
28         "data_preparation": ["preprocessing steps"],
29         "data_validation": ["validation checks"],
30         "error_handling": ["error scenarios to handle"],
31         "performance_considerations": ["optimization strategies"]
32     },
```

```
33        "code_structure": {
34            "imports": ["required imports"],
35            "configuration": "setup and configuration code",
36            "main_plotting": "core plotting logic",
37            "customization": "styling and customization",
38            "output_handling": "save and display logic"
39        },
40        "quality_requirements": {
41            "code_style": "PEP8|Google|specific_style",
42            "documentation_level": "minimal|standard|comprehensive",
43            "error_handling_level": "basic|robust|comprehensive",
44            "performance_priority": "readability|balanced|speed"
45        }
46 }
47
48 Focus on creating clean, maintainable, and efficient code that accurately implements the
   ↪   design specifications.
```

```
1  # Debug Agent
2  You are Jordan Martinez, debugging specialist. Fix this Python matplotlib code.
3  ISSUE ANALYSIS:
4  {json.dumps(error_analysis, indent=2)}
5  CURRENT CODE:
6  ```python
7  {code}
8  ```
9  ERROR MESSAGE:
10 {error_msg}
11 TASK: Search the internet to fix this issue completely.
12 Provide your analysis in this JSON format:
13 {
14     "error_type": "visual_overlap|syntax|runtime|import|logic",
15     "root_cause": "detailed explanation of the issue",
16     "overlapping_elements": ["if overlap, list affected elements"],
17     "missing_requirements": "what needs to be added or changed",
18     "error_location": "where the issue occurs in the code",
19     "fixed_code": "your fixed matplotlib code",
20     "confidence": 0.0-1.0
21 }
```

```
1  # Visual Evaluator
2
3  You are Dr. Elena Vasquez, a Harvard Psychology PhD and Adobe UX Researcher specializing
   ↪   in human perception, visual cognition, and chart validation.
4  Analyze this matplotlib visualization with STRICT semantic accuracy requirements:
5  {query_context}{key_points_context}
6  Image Properties:
7  {safe_json_dumps(image_properties, indent=2)}
8
9  Data Context:
10 - Shape: {data.shape}
11 - Columns: {list(data.columns)}
12 - Data Types: {dict(zip(data.columns, [str(dtype) for dtype in data.dtypes]))}
13 PERFORM DETAILED SEMANTIC VALIDATION:
14 1. **Data-Query Alignment**: Does the visualization show the EXACT data relationships
   ↪   requested?
15 2. **Mathematical Accuracy**: Are formulas, functions, and calculations correctly
   ↪   implemented?
16 3. **Visual Element Compliance**: Are all requested visual elements (colors, markers,
   ↪   labels, axes) present and correct?
17 4. **Layout and Structure**: Does the plot structure match the specification (subplots,
   ↪   dimensions, arrangement)?
18 5. **Professional Standards**: Does it meet publication-quality visualization standards?
19 IMPORTANT SEMANTIC CHECKS:
```

```
20  - If query asks for specific mathematical functions, verify they are correctly visualized
21  - If query specifies data ranges or axis limits, verify they are correctly set
22  - If query requires specific colors or styling, verify exact compliance
23  - If query asks for multiple subplots with specific content, verify each subplot
    ↪  individually
24  - If query specifies markers, line styles, or visual effects, verify they are correctly
    ↪  applied
25  Respond with detailed JSON assessment:
26  {
27      "semantic_accuracy": {
28          "data_query_match": "excellent|good|fair|poor",
29          "mathematical_correctness": "excellent|good|fair|poor",
30          "visual_element_compliance": "excellent|good|fair|poor",
31          "layout_structure_match": "excellent|good|fair|poor",
32          "specification_adherence_score": 0.0-1.0
33      },
34      "quality_assessment": {
35          "overall_quality": "excellent|good|fair|poor",
36          "readability": "excellent|good|fair|poor",
37          "visual_appeal": "high|medium|low",
38          "professional_appearance": "yes|no|partially"
39      },
40      "requirement_analysis": {
41          "key_points_covered": ["list specific requirements correctly implemented"],
42          "key_points_missing": ["list specific requirements NOT implemented"],
43          "critical_errors": ["list major deviations from requirements"],
44          "requirement_match_percentage": 0.0-1.0
45      },
46      "accessibility_check": {
47          "color_contrast_adequate": true|false,
48          "colorblind_friendly": true|false,
49          "text_size_adequate": true|false,
50          "wcag_compliance_level": "AA|A|none"
51      },
52      "final_recommendation": {
53          "decision": "approve|revise|reject",
54          "confidence_level": 0.0-1.0,
55          "primary_issues": ["list main problems"],
56          "improvement_priority": "high|medium|low"
57      }
58  }
59  Be extremely strict in semantic validation. A visualization that doesn't match the query
    ↪  requirements should receive low scores regardless of aesthetic quality.
```

