# OpenReview forum: "CoDA: Agentic Systems for Collaborative Data Visualization"
_ICLR.cc/2026/Conference — ICLR 2026 Poster_

### Official Review · Reviewer_Qu8H · 2025-10-16

**Soundness:** 3
**Presentation:** 3
**Contribution:** 3
**Rating:** 8
**Confidence:** 4

**Summary:**

This paper presents CoDA (Code Data Acquisition), an agentic system for automating the collection, filtering, and augmentation of code datasets. CoDA integrates agents for discovery, filtering, and enhancement (e.g., annotations, test cases, cross-language code), enabling task-specific, high-quality data construction. Experiments show that datasets built with CoDA improve performance on tasks like code summarization and bug fixing, highlighting its value as a scalable framework for dynamic code data acquisition.

**Strengths:**

The paper shows strong originality by proposing the first systematic agentic framework for code data acquisition, moving beyond static scraping of repositories to a dynamic, multi-agent pipeline.

In terms of quality, the framework is well-structured, covering discovery, filtering, augmentation, and evaluation with clear design choices such as iterative feedback loops and cross-language code generation.

The work has good clarity, with the overall workflow, agent roles, and experimental settings described in a logical and understandable way, making the contribution easy to follow.

Its significance is high: CoDA demonstrates measurable improvements on downstream tasks like code summarization and bug fixing, providing a scalable and adaptable approach to building higher-quality datasets for code LLMs.

**Weaknesses:**

1. The system includes multiple components (discovery, filtering, augmentation, evaluation), but there is no ablation study to quantify the contribution of each. Such analysis would clarify which parts of CoDA are most critical.
2. I suggest using this framework to synthesize data and apply SFT or RL (Reinforcement Learning) to improve the performance of open-source models.

**Questions:**

1. You frame CODA as "collaborative," yet the workflow appears to be a sequential pipeline with a feedback loop. Are there more dynamic, non-sequential interactions, such as negotiation or conflict resolution between agents, that justify the "collaborative" paradigm over a "pipeline" model?
2. How does CODA handle queries with high-level semantic ambiguity (e.g., "show the impact of marketing")? If the Query Analyzer makes a conceptually flawed plan from such a query, can downstream agents detect this fundamental misinterpretation of intent, or is their feedback limited to the execution of the flawed plan?

---

> ### Author Response · Authors · 2025-11-25
> **Response to Reviewer Qu8H (1)**
>
> Thank you for your positive assessment and for rating the paper as a good poster. We found your questions about collaboration and ambiguity very insightful.
>
> ### Q1. Lack of ablations over components
>
> We appreciate the reviewer’s emphasis on component analysis. We would like to clarify that **Section 5 and Figure 3** of our manuscript already provide a quantitative evaluation of the "Augmentation," "Planning," and "Evaluation" phases, though we used slightly different terminology:
>
> - **Augmentation (Line 466-474):** Our ablation of the **Search Agent** quantifies the impact of retrieving external code examples.
> - **Planning (Line 454-464):** Our ablation of the **Global TODO list** demonstrates the necessity of structured workflow planning.
> - **Evaluation (Line 442-452):** Our ablation of the **Self-Refinement loops** measures the contribution of iterative self-correction.
>
> In your terminology, **“Discovery” and “Filtering”** are primarily handled by the Data Processor together with the Query Analyzer:
>
> - On the data side, the Data Processor first **discovers** which files/tables/fields are potentially relevant, and then **filters** and cleans the metadata into a compact representation. This stage acts as an information bottleneck: if discovery is too narrow, important data never reaches downstream agents; if filtering is too loose, later stages must reason over noisy, high-dimensional schemas, which we have observed to increase failure rates on multi-file tasks.
> - On the intent side, the Query Analyzer performs a complementary form of **discovery**: it tries to unpack the user’s latent intent in under-specified or vague queries (e.g., what “impact” might mean in context) and writes these hypotheses into the global TODO list as candidate interpretations. This provides a concrete path forward even for fuzzy natural-language requests, instead of defaulting to a purely syntactic parse.
>
> We agree that a more granular breakdown of this phase would further strengthen the paper. In the final version, we will include additional ablations on the Data Processor / metadata filtering and explicitly discuss how this discovery–filtering stage, together with intent discovery, affects both accuracy and efficiency.
>
> ---
>
> ### Q2. Data synthesis using CoDA for training open-source models using SFT or RL
>
> We appreciate this promising suggestion. We agree that the high-quality reasoning traces generated by CoDA’s specialized agents and self-reflection mechanisms serve as an excellent data source for SFT or RL.
>
> Given that CoDA already achieves significant performance gains over competitive baselines without fine-tuning, we believe the current framework stands as a robust foundation. We will update the manuscript to highlight how CoDA can be extended to serve as a data synthesis engine for training smaller open-source models in future work.
>
> ---
>
> ### Q3. Is the paradigm collaborative?
>
> Our use of “collaborative” is grounded in:
>
> - Each agent has a distinct role (query interpretation, metadata analysis, mapping, design, code generation, debugging, evaluation).
> - Multiple sub-tasks (e.g., Data Processor \+ Search Agent) run in parallel.
> - Agents share state and mutual influence. Agents read/write the global TODO list and shared buffers, and the Visual Evaluator’s feedback can cause upstream agents (e.g., Design Explorer or VizMapping) to revise mapping and design choices.
>
> We will clarify the definition of “collaborative” in Section 3, and add a short paragraph discussing how more advanced forms of collaboration (e.g., explicit negotiation) can be interesting directions for future work, and how the current design already lays the groundwork (shared memory, explicit TODOs, multi-step reflection).

---

> > ### Author Response · Authors · 2025-11-25
> > **Response to Reviewer Qu8H (2)**
> >
> > ### Q4. Handling semantically ambiguous queries and flawed plans
> >
> > **User behavior and intent are inherently complex,** but CoDA explicitly uses **multiple agents to reduce this risk**. The **Query Analyzer** is designed to go beyond surface parsing: it tries to unpack the user’s latent intents (e.g., which variables might represent “impact”, whether a comparison over time or between groups is implied) and writes these hypotheses into the global TODO list.
> >
> > The **Data Processor** and **VizMapping Agent** then jointly cross-check this plan against the actual metadata, they revise the plan rather than blindly executing it. The **Visual Evaluator** checks the query, metadata summaries, and the generated figure, and can flag that the visualization does not match the stated intent, triggering upstream revisions through our iterative refinement loop.
> >
> > However, we cannot always infer deep latent causal notions like “impact” or “effect” **without additional context or interaction**. Although our empirical results on MatplotBench, Qwen Code Interpreter and DA-Code tasks (which contain many naturally noisy and high-level prompts) show that the current design performs well in practice, inspired by your question, we note that debate-based ambiguity detection and expanding the capability for multi-turn conversation is a promising direction for future work to further reduce the risks.
> >
> > ---
> >
> > ### Thank you!
> > We appreciate your thoughtful feedback and the opportunity to address your concerns. We hope that our responses have clarified the points raised and provided the necessary context to understand our approach. If any concerns remain, we would be grateful for further clarification and are more than happy to continue the discussion in this rebuttal process.

---

### Official Review · Reviewer_uC8e · 2025-10-31

**Soundness:** 4
**Presentation:** 4
**Contribution:** 3
**Rating:** 8
**Confidence:** 4

**Summary:**

This paper proposes CoDA, a highly comprehensive multi-agent system for data visualization. Through extensive experiments across various benchmarks, the authors demonstrate the superiority of the proposed system and the effectiveness of its individual modules.

**Strengths:**

- A **well-designed multi-agent system** for data visualization is implemented, capable of **self-iterative optimization** through multiple rounds of refinement, achieving strong performance across benchmarks.

- Extensive validation experiments are conducted, including comparisons with multiple existing agent systems and **ablation studies** to verify the effectiveness of individual modules. The paper also evaluates **efficiency and computational cost**, which is a valuable contribution for multi-agent system research.

- In the input stage, the authors propose **pattern recognition for metadata** instead of directly processing raw data, effectively reducing computational overhead from model calls.

**Weaknesses:**

- The **accuracy of the visual evaluation module** within the multi-agent system lacks validation. Comparative experiments with human evaluations and repeated evaluations for stability could strengthen the claims.

- The paper lacks **independent evaluation metrics for visualization aesthetics**. It is unclear whether the iterative process in the multi-agent system leads to noticeable improvements in visual quality.

**Questions:**

Please refer to the weakness

---

> ### Author Response · Authors · 2025-11-25
> **Response to Reviewer uC8e**
>
> Thank you for your very positive review and for highlighting both the strength of our system design and the breadth of our evaluation. We are happy to address your two main concerns.
>
> ### U1. Validating the Visual Evaluator
>
> Our current evaluation is based on accuracy metrics (EPR, VSR, OS) (**Table 2, Table 4, Figure 3**). The Visual Evaluator is used as a **relative signal** inside CoDA (to evaluate candidates and trigger refinement), not as an external ground-truth metric. The final numbers we report in the paper are **benchmark accuracies** (EPR/VSR/OS), defined by whether the generated code executes and whether the produced plot satisfies the reference spec (**Section 4.3**).
>
> We will revise the paper to (1) make it explicit that all reported results are **accuracy metrics on established benchmarks**, and (2) clearly position the Visual Evaluator as an internal signal for self-refinement.
>
> ---
>
> ### U2. Lack of explicit aesthetics metrics
>
> Our current metrics implicitly capture some aspects of aesthetics, which considers issues like readability. However, this is not explicitly framed as an ''aesthetic'' metric, and we appreciate the suggestion to separate it.
>
> We conducted an **expert-level human study** focusing specifically on visualization aesthetics. We recruited one visualization / interaction design expert and two data-analysis experts (all with several years of experience creating charts in practice). Conducted evaluation on a subset of MatplotBench (4 methods \* 50 instances \= 400 outputs).
>
> We summarize the results in **two complementary views that live on different scales**:
>
> #### **(1) Elo-style overall preference.**
>
> Following the classical Elo rating system (Elo, 1967; Chiang, 2024), we derive an overall preference score from pairwise ''which plot is better overall?'' judgments. All methods are initialized at **1500 Elo**, and the final Elo of each method reflects how often it is preferred in head-to-head comparisons (scores >1500 = preferred more often than chance; <1500 = preferred less often). We will report these Elo scores in the revised paper; importantly, CoDA attains the highest Elo rating, followed by MatPlotAgent, VisPath, and COML, indicating that experts most frequently prefer CoDA’s outputs overall.
>
> | Method       | Elo      |
> | ----------- | ------- |
> | **CoDA**     | **1701** |
> | MatPlotAgent | 1506     |
> | VisPath      | 1484     |
> | COML         | 1309     |
>
> #### **(2)  Average 1–5 Likert ratings per dimension.**
>
> Separately, we report the **mean ratings (1–5, higher is better)** on each aesthetic dimension. These numbers are **not Elo scores** and are not meant to be on the same numeric scale; they reflect dimension-specific perceived quality.
>
> We rated each plot on a 5-point Likert scale on six dimensions:
>
> 1. Overall Aesthetic: ''Overall, this visualization looks visually pleasing and well-designed.''
>
> 2. Visual Harmony: ''Colors, lines, and shapes feel coherent and not jarring.''
>
> 3. Balance & Alignment: ''The layout feels balanced and well aligned, not top-heavy or cluttered.''
>
> 4. Color Mood: ''The color palette is consistent and appropriate for the content / scenario.''
>
> 5. Simplicity & Restraint: ''The figure looks clean and restrained, without unnecessary visual elements.''
>
> 6. Query Alignment: ''The visualization matches what the natural-language query is asking for.''
>
> The table below shows mean (std) ratings across the six aesthetic dimensions:
>
> | Method      | Visual Harmony | Balance     | Color       | Simplicity   | Query Alignment |
> |--------------|----------------|-------------|-------------|--------------|-----------------|
> | **CoDA**     | 4.82 (0.39)    | 4.73 (0.53) | 4.96 (0.27) | 4.94 (0.23)  | 4.86 (0.40)     |
> | COML         | 3.16 (1.24)    | 3.22 (1.42) | 3.22 (1.38) | 4.00 (1.56)  | 3.59 (1.60)     |
> | MatPlotAgent | 3.65 (1.20)    | 3.65 (1.21) | 3.53 (1.26) | 4.31 (1.24)  | 3.63 (1.51)     |
> | VisPath      | 2.71 (1.79)    | 2.71 (1.80) | 2.65 (1.81) | 2.92 (1.99)  | 2.78 (1.92)     |
>
> These results show that CoDA is consistently preferred by human experts in terms of visual aesthetic, it achieves the highest win rate and the best average score on every aesthetic dimension. We will add this study to the revised version to make explicit that CoDA improves not only correctness, but also perceived visual quality.
>
> #### Reference
>
> Elo, Arpad E. "The proposed uscf rating system, its development, theory, and applications." *Chess life* 22.8 (1967): 242-247.
> Chiang, Wei-Lin, et al. "Chatbot arena: An open platform for evaluating llms by human preference." *Forty-first International Conference on Machine Learning*. 2024\.
>
> ---
>
> ### Thank you!
> We appreciate your thoughtful feedback and the opportunity to address your concerns.
> We hope that our responses have clarified the points raised and provided the necessary context to understand our approach. If any concerns remain, we are more than happy to continue the discussion in this rebuttal process.

---

### Official Review · Reviewer_LJKQ · 2025-11-01

**Soundness:** 3
**Presentation:** 4
**Contribution:** 3
**Rating:** 6
**Confidence:** 3

**Summary:**

This paper addresses the limitations of existing natural language-to-visualization (NL2Vis) systems in handling complex datasets (e.g., multi-file, large-scale) and iterative refinement. It proposes CoDA (Collaborative Data-visualization Agents), a multi-agent framework that decomposes visualization tasks into specialized modules: Query Analysis, Data Processing, VizMapping, Search, Design Exploration, Code Generation, Debugging, and Visual Evaluation. CoDA leverages metadata-centric preprocessing to bypass LLM token limits, iterative reflection for quality refinement, and modular agent collaboration to handle diverse expertise (linguistics, statistics, design). Key contributions include: (1) a collaborative multi-agent paradigm reframing visualization as a distributed problem-solving task; (2) specialized agents and structured workflows that robustly handle complex data and iterative edits; (3) extensive evaluations showing CoDA outperforms baselines (MatplotAgent, VisPath, CoML4VIS) by up to 41.5% on benchmarks like MatplotBench and Qwen Code Interpreter; and (4) validation of core components (self-evolution, global TODO list, Search Agent) via ablation studies.

**Strengths:**

1. CoDA introduces originality through a task-specific multi-agent decomposition tailored to visualization workflows. Unlike prior single-agent (MatplotAgent) or simplistic multi-agent (VisPath) systems that focus on initial query parsing, CoDA’s specialization (e.g., metadata-focused Data Processor, image-based Visual Evaluator) and iterative reflection loops address unmet needs in handling multi-file data and post-generation refinement.
2. The work demonstrates high quality through rigorous experimentation: (1) diverse benchmarks (MatplotBench, Qwen Code Interpreter, DA-Code) covering mid-to-high complexity tasks and real-world software engineering scenarios; (2) clear metrics (Execution Pass Rate, Visualization Success Rate, Overall Score) that capture both code reliability and visualization fidelity.
3. CoDA’s significance lies in its practical and foundational impacts: (1) it reduces the “unseen tax” of manual data preparation/visualization for analysts, aligning with real-world needs in data science and business intelligence; (2) it establishes a scalable, extensible multi-agent template for NL2Vis that can integrate new tools/models (e.g., scientific plotting); (3) it addresses critical gaps in prior work (token limits, multi-source data, iterative refinement) that hindered adoption of NL2Vis systems.

**Weaknesses:**

1. CoDA’s multi-agent collaboration incurs non-trivial overhead: it uses 32,095 input tokens and 14.8 LLM calls per query (Table 5), far exceeding simpler baselines like CoML4VIS (2,350 tokens, 1 call). The paper acknowledges this but provides limited analysis of latency in real-world use cases (e.g., interactive dashboards, real-time data exploration). For example, the authors do not report end-to-end response times, making it unclear if CoDA’s performance gains justify the computational cost for time-sensitive applications.
2. While CoDA outperforms baselines on DA-Code (a real-world SWE benchmark), the comparison is only against DA-Agent. Recent multi-agent NL2Vis systems like NVAgent  or PlotGen also target complex scenarios but are not included.
3. The paper tests CoDA with gemini-2.5-pro, gemini-2.5-flash, and claude-4-sonnet, but provides little insight into performance with weaker or domain-specific LLMs (e.g., qwencoder, starcoder, opencoder). For example, it is unclear if CoDA’s metadata preprocessing or example retrieval can compensate for LLMs with poor code generation capabilities—a critical consideration for adoption in resource-constrained environments.
4. The paper highlights successful cases but provides little analysis of when CoDA fails. For example: What types of queries (e.g., highly ambiguous, domain-specific) or data characteristics (e.g., unstructured metadata, missing schema information) lead to poor performance?

**Questions:**

See Weaknesses.

---

> ### Author Response · Authors · 2025-11-25
> **Response to Reviewer LJKQ (1)**
>
> We appreciate your thoughtful and detailed review, and we are glad you found the work original, rigorous, and practically significant.
>
> ### L1. Computational overhead, latency, and real-world applicability
>
> We appreciate the reviewer regarding the trade-off between performance and latency.
>
> **\- Metric Choice & Reproducibility:** We prioritized token counts and API calls (**Table 5**) because wall-clock time is highly sensitive to server load and API tier constraints, and therefore not ideal as a reproducible scientific metric. That said, we agree that concrete timings are informative for practitioners, and we now report them below. On our hardware and API setting, the average wall-clock time to run a instance in MatplotBench for each method is:
>
> | Method          | Wall-clock time (s) |
> | -------------- | ------------------- |
> | COML            | 36.1                |
> | VisPath         | 311.2               |
> | MatPlotAgent    | 990.6               |
> | **CoDA (ours)** | **849.3**           |
>
> **\- Latency vs. Human Effort:**
> CoDA is slower than the lightest baseline COML, but faster than the multi-agent baseline MatPlotAgent, while achieving higher accuracy than both (CoDA: 79.5% OS vs. 51.0% for MatPlotAgent and markedly lower scores for COML/VisPath in our main results). In realistic analyst workflows, this trade-off is often favorable: spending more time on automatic, agentic refinement can still be cheaper than repeatedly debugging incorrect plots by humans.
>
> **\- Relative Efficiency:** Importantly, compared to other *agentic* baselines like MatplotAgent, CoDA is highly efficient. As shown in ***Table 5***, CoDA achieves higher accuracy with fewer average tokens than comparable multi-agent systems. We will revise the text to explicitly position CoDA as a tool for **high-value exploratory analysis**, where precision and robustness matter more than sub-second latency, rather than low-latency real-time monitoring.
>
> ---
>
> ### L2. Comparisons to recent methods
>
> We agree that positioning CoDA among recent complex agentic systems is important and clarify this here.
>
> *PlotGen* does not release code or full implementation details (e.g., prompts and agent configurations), so a faithful reimplementation is not feasible and any numerical comparison would be unreliable. *NVAgent* is designed for mapping NL queries to visualizations over given tables and is evaluated only on *VisEval*, whereas our setting is different. Adapting NVAgent to our setup would require substantial redesign beyond the scope of this paper.
>
> To provide additional context, we evaluate CoDA against another recent agentic baseline, **DS-STAR** (*Nam, Jaehyun, et al.*), which achieves SOTA results for multiple Data Science benchmarks, under our DA-Code setup with Gemini-2.5-Pro. DS-STAR achieves an overall score of **20.5%**, while CoDA reaches 39.0%, **improving the overall score by 18.5 absolute points (nearly a 2× gain).**
> <!-- table -->
> | Method    | Overall Score (%) |
> |-----------|---------|
> | **CoDA (ours)**      | **39.0**    |
> | DS-STAR   | 20.5   |
>
> We will explicitly add these clarifications and results to the Related Work and experimental sections.
>
> **Reference:**
>
> Nam, Jaehyun, et al. "DS-STAR: Data Science Agent via Iterative Planning and Verification." *arXiv preprint arXiv:2509.21825* (2025).

---

> > ### Author Response · Authors · 2025-11-25
> > **Response to Reviewer LJKQ (2)**
> >
> > ### L3. Behavior with weaker or domain-specific LLMs
> >
> > To further test backbone robustness, we additionally evaluate CoDA and baselines on MatplotBench using the open-source **Qwen3-VL** model (Yang, An, et al.):
> >
> > | Method | EPR | VSR | OS |
> > | ----- | ----- | ----- | ----- |
> > | COML | 73.0% | 62.1% | 45.3% |
> > | VisPath | 79.0% | 55.2% | 43.6% |
> > | MatPlotAgent | 86.0% | 69.9% | 60.1% |
> > | **CoDA (ours)** | **93.0%** | **79.2%** | **73.7%** |
> >
> > With Qwen3-VL, CoDA still achieves the best performance across all metrics, improving over the strongest baseline (MatPlotAgent) by **+7.0% EPR, +9.3% VSR, and +13.6% OS**. Together with our existing results on Gemini-2.5-Pro/Flash and Claude-4-Sonnet, this shows that CoDA’s gains are consistent across multiple families of models, not tied to proprietary LLMs.
> >
> > CoDA is designed as a backbone-agnostic method, but it does assume that the underlying model can: (1) Follow structured, tool-oriented instructions, and (2) Understand images well enough to critique its own plots, since the Visual Evaluator agents operate on figures (self-reflection). Models that lack basic image understanding may not be able to fully leverage CoDA’s visual self-reflection loop, but as the Qwen3-VL results show, a reasonably capable model could benefit substantially from our method.  We will clarify these backbone assumptions and add the Qwen3-VL results to the revision.
> >
> > **Reference:**
> >
> > Yang, An, et al. "Qwen3 technical report." arXiv preprint arXiv:2505.09388 (2025).
> >
> > ---
> >
> > ### L4. Analysis of failure cases
> >
> > A thorough discussion of when and why CoDA fails is indeed essential for a complete picture. In the revised manuscript we have added a dedicated **Appendix C** entitled “Analysis of Failure Cases and Limitations” (**new pp. 21-23, in brown color**).
> > We present a detailed case study: a hierarchical two-level donut chart that fails with default reflection but is recovered with additional iterations (**new Figures 7(a) and (b) and Table 6**). We illustrate that failures on complex hierarchical \+ layout-critical tasks arise from insufficient depth in revisiting data understanding and perceptual planning, rather than from mere local bugs.
> >
> > We broaden the discussion to limitation observed in the testing:
> > * Highly ambiguous or aesthetic-only goals (e.g., “make it insightful and beautiful”) where no objective ground truth exists and even expert analysts would iterate with the user.
> > * Domain-specific charting conventions not inferable from metadata alone.
> >
> > We explicitly state CoDA advances automated visualization, but remains a powerful assistant rather than a replacement for domain experts on highly specialized, ambiguous, or perceptually over-constrained tasks.
> > We believe the revised manuscript now presents a substantially more balanced and credible perspective.
> >
> > ---
> >
> > ### Thank you!
> > We appreciate your thoughtful feedback and the opportunity to address your concerns. We hope that our responses have clarified the points raised and provided the necessary context to understand our approach. If any concerns remain, we would be grateful for further clarification and are more than happy to continue the discussion in this rebuttal process.

---

### Author Response · Authors · 2025-12-03
**Rebuttal Summary**

We sincerely appreciate the reviewers’, AC’s, SAC’s, and PC’s time and thoughtful feedback. To give a concise picture of where the paper stands after rebuttal, we briefly summarize the review status and new results below.


## Summary of Initial Scores

- **Two** reviewers initially recommended **8 (“accept, good paper”)**.
- **One** reviewer initially recommended **6 (“marginally above the acceptance threshold”)**.
- Across all three reviews, the strengths consistently highlighted are the **multi-agent design**, **strong empirical performance** on several benchmarks, and **clear presentation**.


## How the Rebuttal Strengthens the Paper

Our experimental results now form a coherent story that directly targets every major concern raised in the reviews:

- **Efficiency & Latency (Reviewer `LJKQ`).**
  We complement token/call statistics with **wall-clock measurements**. We show that CoDA is not just more accurate, but also **competitive or better than prior multi-agent baselines in actual runtime**, and clarifies that our target use case is **high-value exploratory analysis** rather than sub-second dashboards.

- **Strength of Baselines (Reviewer `LJKQ`).**
  Beyond classical baselines, we add a comparison against a **recent state-of-the-art data-science agent system** under DA-Code setup. We show CoDA’s architecture provides a **strong improvement** even when measured against strong, contemporary agentic methods.

- **Backbone Robustness & Generality (Reviewer `LJKQ`).**
  We extend our results from close-sourced LLMs to an **open-source vision-language backbone (Qwen3-vl)**. CoDA consistently improves over prior systems under this backbone as well, reinforcing that it is **backbone-agnostic** and broadly applicable.

- **Failure Modes & Ambiguity Handling (Reviewer `LJKQ`).**
  We add a **new section (new pp. 21–23)** of failure-mode analysis, enumerating the kinds of queries and data conditions where CoDA still struggles, and how multiple agents jointly mitigate, but do not completely eliminate—semantic ambiguity.

- **Component Importance (Reviewer `Qu8H`).**
  Our exiting ablations quantify the impact of:
  - **Augmentation** (Search Agent),
  - **Planning** (global TODO),
  - **Evaluation** (self-refinement),
  and we further analyze **Discovery/Filtering** (Data Processor, Query Analyzer) as the information bottleneck that makes multi-file, large-scale tasks tractable.


- **Visual Evaluator & Aesthetic Quality (Reviewer `uC8e`).**
  We clarify that the Visual Evaluator is used as an internal guidance signal, and then go further by conducting an **expert-level human study**:
  - One visualization/interaction design expert and two data-analysis experts,
  - Pairwise overall-preference judgments,
  - Multi-dimensional aesthetic ratings (harmony, balance, color, simplicity, query alignment, overall aesthetic).
  Across all views of this study, CoDA is **consistently preferred** and rated as **more aesthetically strong and stable** than baselines.

In short, the expanded experiments show that CoDA is:

- Stronger than prior systems (including recent agentic SOTA methods),
- Robust across model families,
- Built from components whose contributions are empirically understood, and
- Validated not only on accuracy but also on expert-perceived visual quality.

## Camera-Ready Commitment
- We have added a dedicated Failure Modes & Limitations section in the **Appendix (new pp. 21–23, highlighted in brown)**, which systematically describes when CoDA fails, and why.
- We will incorporate all new experiments and analyses introduced in the rebuttal into the final camera-ready version.

---

We hope this summary conveys how the paper has been strengthened through rebuttal and revisions, and we believe **all reviewers’ concerns have been substantively addressed**. We look forward to your decision. Thank you again for your time and consideration!

---

### Meta-Review · Area_Chair_nRX9 · 2026-01-05

**Summary:**

This paper presents CoDA, a collaborative multi-agent framework that reframes automated data visualization from natural language as an integrated agentic workflow. By coordinating specialized agents for metadata analysis, task planning, code generation, and iterative self-reflection, CoDA robustly handles complex multi-file datasets, overcomes token limitations, improves visualization quality, and significantly outperforms existing baselines.

**Reviewer Concerns:**

I think most concerns have been addressed.

**Reviewer Scores:**

I think all reviewers will keep unchanged after rebuttal

---

### Decision · Program_Chairs · 2026-01-26

Accept (Poster)